# MoLE 🦫: Human-centric Text-to-image Diffusion with Mixture of Low-rank Experts

## Abstract

Text-to-image diffusion has attracted vast attention due to its impressive image-generation capabilities. However, when it comes to human-centric text-to-image generation, particularly in the context of faces and hands, the results often fall short of naturalness due to insufficient training priors. We alleviate the issue in this work from two perspectives. **1)** From the data aspect, we carefully collect a *human-centric dataset* comprising approximately one million high-quality human-in-the-scene images and two specific sets of close-up images of faces and hands. These datasets collectively provide a rich prior knowledge base to enhance the human-centric image generation capabilities of the diffusion model. **2)** On the methodological front, we propose a simple yet effective method called **M**ixture **o**f **L**ow-rank **E**xperts (**MoLE**) by considering low-rank modules trained on close-up hand and face images respectively as experts. This concept draws inspiration from our observation of low-rank refinement, where a low-rank module trained by a customized close-up dataset has the potential to enhance the corresponding image part when applied at an appropriate scale. To validate the superiority of MoLE in the context of human-centric image generation compared to state-of-the-art, we construct two benchmarks and perform evaluations with diverse metrics and human studies. More visualization, datasets, models, and code will be released on our anonymous webpage.

## 1 Introduction

Text-to-image Stable Diffusion (Rombach et al., 2022) has recently gained great attention due to its impressive capability to generate plausible images that align with the textual semantics provided. Its open source further boosts the development within the community and fosters a prosperous trend for artificial intelligence generated content (AIGC) with substantial progress achieved. However, there remains a challenge in "human-centric" text-to-image generation — current models encounter issues with producing natural-looking results, particularly in the context of faces and hands[1], which has numerous important real-world applications, *e.g.*, business posters, virtual reality, *etc*.

This issue is demonstrated in the leftmost column (without LoRA) of Fig 1, that, Stable Diffusion struggles to produce a realistic human-centric image with accurate facial features and hands. We dive deep into this issue and find two factors that potentially contribute to this problem. Firstly, the absence of *comprehensive* and *high-quality* human-centric data [2] within the training dataset LAION5B (Schuhmann et al., 2022) makes diffusion models lack sufficient human-centric prior; Secondly, in the human-centric context, faces and hands represent the two most complicated parts due to high variability, making them challenging to be generated naturally.

We alleviate this problem from two perspectives. On one hand, we collect a human-in-the-scene dataset of high-quality and high-resolution from the Internet. Basically, the resolution of each image

---

[1]The Hugging Face website acknowledges that "Faces and people in general may not be generated properly." See limitation in `https://huggingface.co/runwayml/stable-diffusion-v1-5`.

[2]We randomly sample 35w human-centric images from LAION2b-en and find that the average height and width are 455 and 415 respectively. Most of them are between 320 and 357. In sampled 1000 images, we also find little (almost zero) high-quality close-ups of face and hand (especially hand). This makes LAION2b-en limitd in providing comprehensive human-centric knowledge.

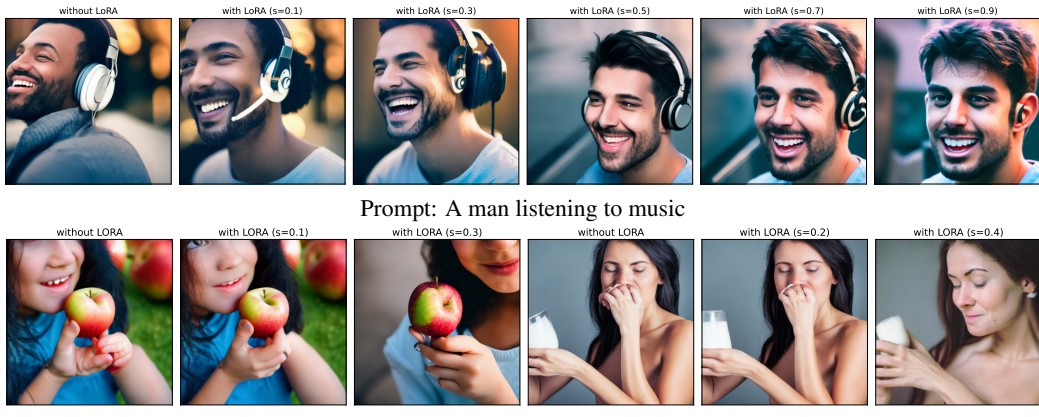

Prompt: A man listening to music

Prompt: A girl eating an apple         Prompt: A woman drinking milk

Figure 1: **Motivation of Mixture of Low-rank Experts.** The first row is produced by low-rank module trained on the face dataset. The second row is produced by low-rank module trained by hand dataset. We observe that low-rank module is capable of refining the corresponding part with a proper scale weight.

is various and over $1024 \times 2048$. The dataset covers different races, various human gestures, and activities. We carefully process each image to preserve sufficient details and resize it $512 \times 512$ for training. Moreover, considering the limitations of Stable Diffusion in producing natural face and hand within a human-centric context, we have taken additional steps to gather two customized close-up datasets that consist of high-quality close-up of both facial and hand regions. However, the close-up of the hand image is limited and less than the amount of face image. To supplement more hand data, we manually crop high-quality hand images from people in images from the previously collected dataset. Finally, we merge all images together and derive a new dataset, which we refer to as *human-centric dataset* containing approximately one million images in total, which we believe can improve the performance of diffusion models in human-centric generation.

On the other hand, during our preliminary experiments, we observe an interesting low-rank refinement phenomenon that inspires us to leverage the idea rooted in mixture of experts (MoE) (Jacobs et al., 1991; Shazeer et al., 2017; Lepikhin et al., 2020; Fedus et al., 2022) to help human-centric image generation. Specifically, we train two low-rank modules (Hu et al., 2021) based on Stable Diffusion v1.5 (SD v1.5) (Rombach et al., 2022) using face and hand datasets respectively [3]. As shown in Fig 1, when combined with a customized low-rank module and using a proper scale weight, SD v1.5 has the potential to refine the corresponding part of the person. Similar results can also be observed in hand in the last row of Fig 1. Thus, in the context of human-centric image generation, intuitively, we could add a certain assignment to adaptively select which specialized low-rank modules to use for a given input and MoE naturally stands out. Additionally, as face and hand often appear simultaneously in an image for a person, motivated by Soft MoE (Puigcerver et al., 2023), we could adopt a soft assignment, allowing multiple experts to handle the input at the same time.

Consequently, considering low-rank modules trained on customized datasets as specialized experts, we propose a simple yet effective method called **M**ixture **o**f **L**ow-rank **E**xperts (**MoLE**). Our method contains three stages: We start by adopting SD v1.5 as a baseline and fine-tune it on our collected human-centric dataset to complement sufficient human-centric prior; We then use two subdatasets, *i.e.*, close-up of face and hand images, to train two low-rank experts separately; Finally, we formulate these two low-rank experts in an MoE form and integrate them with the base model in an adaptive soft assignment manner [4]. To evaluate our method, we construct two human-centric benchmarks using data from COCO Caption (Chen et al., 2015) and DiffusionDB (Wang et al., 2022). The results suggest that MoLE consistently shows superior performance over SD v1.5.

Our contribution can be summarized as:

---

[3]The face data is from Celeb-HQ (Karras et al., 2018). The hand data is from 11k Hands (Afifi, 2019)

[4]Considering that the human face and hand are generally the most frequently observed parts in an image and their bad cases are also extensively discussed or complained in image generation communities, thereby in this work, we primarily focus on the two most important and urgent parts. Our work could also easily involve other human parts, *e.g.*, feet, by collecting a close-up of the feet dataset, training an extra low-rank expert, and accordingly modifying the parameter of the soft assignment.

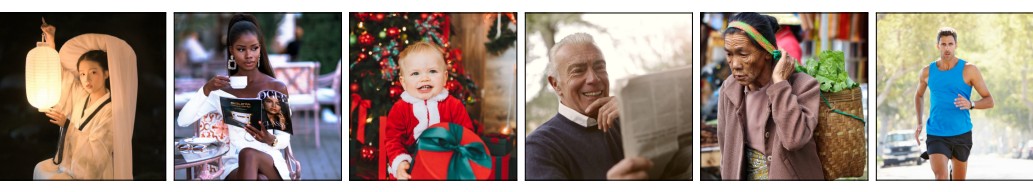

Samples from our human-in-the-scene images

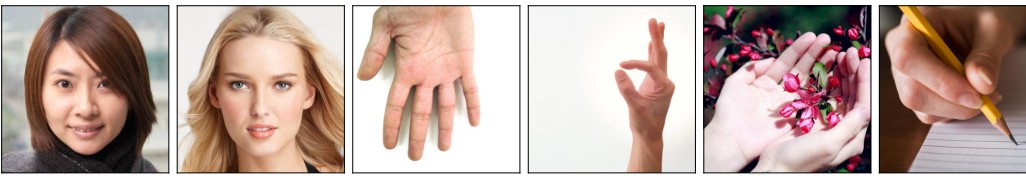

Samples from the close-up of face images          Samples from the close-up of hand images

Figure 2: Some showcases of our human-centric dataset.

- We carefully collect a human-centric dataset comprising around one million high-quality human-centric images. Importantly, we include two specialized datasets containing close-up images of faces and hands to provide a comprehensive human-centric prior.
- We propose a simple yet effective method called Mixture of Low-rank Experts (MoLE) by considering low-rank modules trained on customized hand and face datasets as experts and integrating them into a MoE framework with soft assignment.
- We construct two human-centric evaluation benchmarks from DiffusionDB and COCO Caption. Results show superiority of MoLE over recent state-of-arts in human-centric generation.

## 2 HUMAN-CENTRIC DATASET

Our human-centric dataset contains three parts: human-in-the-scene images, close-up of face images, and close-up of hand images. overall, we have around 1 million high-quality images as shown in Fig 2. Below we introduce each in detail. We put the license and privacy statement in Appendix A.2 to avoid ethical concerns.

### 2.1 HUMAN-CENTRIC DATASET CONSTITUTION

**Human-in-the-scene images.** The human-center images are collected from the Internet. The resolution of these images varies and is basically over $1024 \times 2048$. We primarily focus on high-resolution human-centric images, hoping to supply sufficient priors for diffusion models. These images are diverse w.r.t. occasions, activities, gestures, ages, genders, and racial backgrounds [5]. To enable training, we use a sliding window ($1024 \times 1024$) to crop the image to maintain as much information as possible and then resize cropped parts into $512 \times 512$. For an image, high resolution does not mean high quality. Therefore, we train a VGG19 (Simonyan & Zisserman, 2014) to filter out blurred images. Additionally, considering the crop operation could generate images that are full of background or contain little information about people, we train a VGG19 (Simonyan & Zisserman, 2014) to filter out such bad cases [6]. To ensure the quality, we repeat the two processes multiple times until we do not find any case mentioned above in three times of sampling. By employing these strategies, we can remove amounts of noise and useless images, thereby guaranteeing the image quality.

**Close-up of face images.** The face dataset contains two sources: the first is from Celeb-HQ (Karras et al., 2018) in which we choose images of high quality with size $1024 \times 1024$; The second is from

---

[5]For example, approximately 57.33% individuals identify as white, 14.68% as asian, 9.98% as black, 5.11% as indian, 5.52% as latino hispanic, and 7.38% as middle eastern. Approximately 58.18% are male and 41.82% are female.

[6]In both case, to train the VGG19, we manually collect around 300 positive samples and 300 negative samples as training set, and we also collect around 100 positive samples and 100 negative samples as val set. When training the VGG19, we set batch size to 128, set learning rate to 0.001, and use random flip as data augmentation method. We train the model for 200 epochs and use the best-performing model for subsequent classification. See AppendixA.3 for illustration of negative samples.

| | BLIP2 | ClipCap | MiniGPT-4 | LLaVA |
|---|---|---|---|---|
| | a man in a black jacket leaning against a wall | a man standing in front of a wall with graffiti on it and wearing a black jacket and black shirt | The image shows a young man standing in front of a graffiti-covered brick wall, wearing a black leather jacket and looking off into the distance with a serious expression on his face. | A young man with a black jacket and a black hoodie is standing in front of a graffiti-covered brick wall, looking at the camera with a serious expression. |
| | a little girl is playing in the water | a little girl is playing in the water by a pond with a rock and grass in the background and a waterfall in the foreground | This image shows a small child sitting on a rock in a pond, looking into the water with a curious expression on their face. The child is wearing a brown dress and has dark hair. The pond is surrounded by trees and rocks, and there is a small waterfall in the background. The water in the pond is clear and reflective, creating a mirror effect. The child's reflection can be seen in the water. The overall mood of the image is peaceful and serene. | A young girl in a brown dress squatting near a small pond, looking into the water and admiring her reflection, while a fountain is also present in the background. |

Prompt for MiniGPT-4 and LLaVA:
Describe this image in one sentence with details.

Figure 3: The results of four captioning models. Texts in red are inaccurate descriptions and texts in green are detailed correct descriptions. LLaVA presents a good balance between the level of detail and error rate, and thus is chosen for captioning our dataset.

Flickr-Faces-HQ (FFHQ) (Karras et al., 2019). We sample images covering different skin color, age, sex, and race. We resize them to $512 \times 512$ and there are around 6.4k face images. We do not sample more face images as it is sufficient for low-rank expert training.

**Close-up of hand images.** The hand dataset contains three sources: the first is from 11k Hands (Afifi, 2019) where we randomly sample around 1k images and manually crop them to square; The second is from the Internet where we collect hand images of high quality and resolution with simple backgrounds and use YOLOv5 (Couturier et al., 2021) to detect hands and crop them to $512 \times 512$ with details maintained; The third is from human-in-the-scene images (before processing) where we sample 8k images. We check every image and manually crop the hand of the image to square if the image is appropriate. It is worth noting that in this derived hand dataset, there are abundant hand gestures and interactions with other objects shown in Fig 2, *e.g.*, holding a flower, writing, *etc*. There are 7k high-quality hand images.

## 2.2 IMAGE CAPTION GENERATION

When collecting the dataset, we primarily consider the image quality and resolution and neglect whether it is text paired so as to increase the amount. Thus, producing a caption for each image is required. We investigate four recently proposed SOTA models including BLIP-2 (Li et al., 2023), ClipCap (Mokady et al., 2021), MiniGPT-4 (Zhu et al., 2023), and LLaVA (Liu et al., 2023). We show several cases in Fig 3. One can see that BLIP-2 usually produce simple description and ignore details. ClipCap has a better performance but still lacks sufficient details along with the wrong description. MiniGPT4, although gives detailed descriptions, is inclined to spend a long time (17s on average) generating long and inaccurate captions that exceed the input limit (77 tokens) of the Stable Diffusion CLIP text encoder (Radford et al., 2021). In contrast, LLaVA produces neat descriptions in one sentence with accurate details in a short period (3-5s). Afterward, we manually modify the long LLaVA caption and remove unrelated text patterns, *e.g.*, "The image features that . . . ", "showcasing . . . ", "creating . . . ", "demonstrating . . . ", *etc*. To further ensure the caption alignment of LLaVA, we use CLIP to filter image-text pairs with lower scores.

## 3 METHOD

### 3.1 PRELIMINARY

**Text-conditioned diffusion.** The image generation starts from a random noise $\epsilon$ sampled from Gaussian distribution and gradually becomes a high-fidelity image after multiple iterations of denoising. In each iteration, diffusion models (Ho et al., 2020; Saharia et al., 2022; Rombach et al., 2022; Ramesh et al., 2022; Balaji et al., 2022) predict an estimate of added noise $\tilde{\epsilon}$. In text-guided image generation, diffusion models always perform noise prediction conditioned on input and given text. Eventually, the denoised image is expected to align well with the text in semantics. The training object of a diffusion model $\epsilon_\theta$ is formulated as:

$$\min_{\epsilon_\theta} \mathbb{E}_{\epsilon,t,x,c_p}[\|\epsilon - \epsilon_\theta(x_t, c_p)\|^2], \tag{1}$$

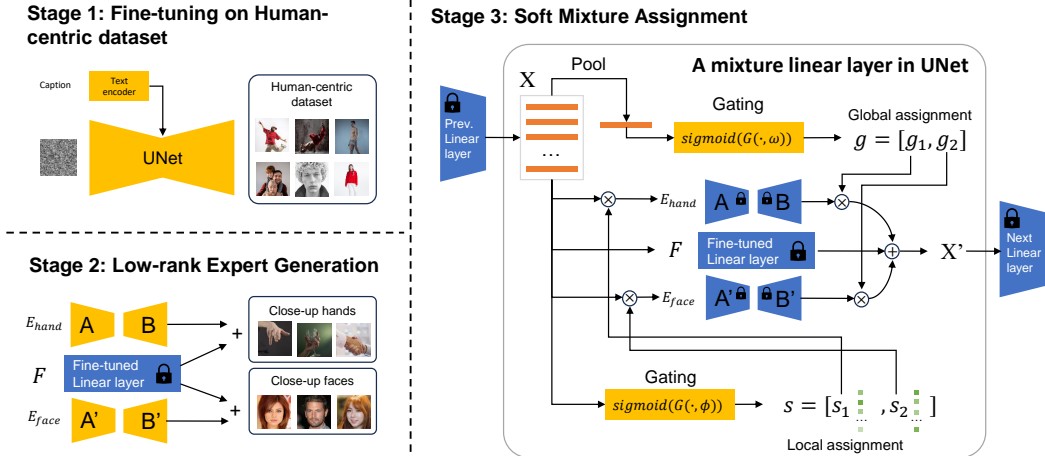

Figure 4: The framework of MoLE. $X$ is the input of any linear layers in UNet. $A$ and $B$ are low-rank matrices.

where $x_t$ is disturbed by applying Gaussian noise $\epsilon \sim N(0, 1)$ in time step $t \sim [1, T]$. $c_p$ is text embedding. The diffusion model $\epsilon_\theta$ is trained to minimize a mean-squared error loss.

To achieve text-guided generation, classifier-free guidance (Ho & Salimans, 2021) is adopted in training where text conditioning $c_p$ is randomly dropped with a fixed probability. This leads to a joint model for unconditional and conditional objectives. When inference, the predicted noise is adjusted to:

$$\tilde{\epsilon}_\theta(x_t, c_p) = \epsilon_\theta(x_t) + s \cdot (\epsilon_\theta(x_t, c_p) - \epsilon_\theta(x_t)), \quad (2)$$

where $s \in (0, 20]$ is guidance scale. Intuitively, the unconditional noise prediction $\epsilon_\theta(x_t)$ is pushed in the direction of the conditioned $\epsilon_\theta(x_t, c_p)$ to yield an image faithful to text prompt. Guidance scale $s$ determines the magnitude of the influence of the text and we set it to 7.5 by default.

**Low-rank Adaptation (LoRA).** Given a customized dataset, instead of training the entire model, LoRA (Hu et al., 2021) is designed to fine-tune the "residual" of the model *i.e.*, $\triangle W$:

$$W^{'} = W + \triangle W \quad (3)$$

where $\triangle W$ is decomposed into low-rank matrices: $\triangle W = AB^T$, $A \in \mathbb{R}^{n \times d}$, $B \in \mathbb{R}^{m \times d}$, $d < n$, $d < m$. During training, we can simply fine-tune $A$ and $B$ instead of $W$, making fine-tuning on customized dataset memory-efficient. In the end, we get a small model as $A$ and $B$ are much smaller than $W$.

**Mixture-of-Experts (MoE).** MoE (Jacobs et al., 1991; Shazeer et al., 2017; Lepikhin et al., 2020; Fedus et al., 2022) is designed to enhance the predictive power of models by combining the expertise of multiple specialized models. Usually, a central "gating" model $G(.)$ selects which specialized model to use for a given input:

$$y = \sum_{i=1} G(x)_i E_i(x). \quad (4)$$

When $G(x)_i = 0$, the corresponding expert $E_i$ will not be activated.

### 3.2    MIXTURE OF LOW-RANK EXPERTS

Motivated by the two potential reasons discussed above that attribute to the poor performance of human-centric generation, our method contains three stages shown in Fig 4. We describe each stage below and put the training details in Appendix A.1.

*Stage 1: Fine-tuning on Human-centric Dataset.* The limited human-centric prior for diffusion model could be caused by the absence of large-scale high-quality human-centric datasets. Considering such a pressing need, our work bridges this gap by providing a carefully collected dataset that contains one million human-centric images of high quality. To fully leverage these images and learn as much prior as possible, we adopt SD v1.5 as baseline and fine-tuning on the dataset. Concretely, we fine-tuning the text encoder and UNet modules (Ronneberger et al., 2015) while fixing the rest parameters. Our ablation study in Sec 4.3.1 shows that this stage is effective and significantly improves performance. The well-trained model is then sent to the next stage.

***Stage 2: Low-rank Expert Generation.*** To construct MoE, in this stage, our goal is to prepare two experts, that are supposed to contain abundant knowledge about the corresponding part. To achieve this, we train two low-rank modules using two customized datasets. One is the close-up face dataset. The other is the close-up hand dataset that contains abundant hand gestures, full details with simple backgrounds, and interactions with other objects. We then use the two datasets to train two low-rank experts with Stable Diffusion trained in stage 1 as base model. The low-rank experts are expected to focus on the generation of face and hand and learn useful context.

***Stage 3: Soft Mixture Assignment.*** This stage is motivated by the low-rank refinement phenomenon in Fig 1 where a specialized low-rank module using a proper scale weight is capable to refine the corresponding part of person. Hence, the key is to activate different low-rank modules with suitable weights. From this view, MoE naturally stands out and we novelly regard a low-rank module trained on a customized dataset, *e.g.*, face dataset, as an expert and formulate in a MoE form. Moreover, for a person, face and hand would appear in an image simultaneously while hard assignment mode in MoE only allows one expert accessible to the given input. Hence, inspired by Soft MoE (Puigcerver et al., 2023), we adopt a soft assignment, allowing multiple experts to handle input simultaneously. Further, considering that the face and hand would be a part of the whole image (local) or occupy the whole image (global), we combine local assignment and global assignment together.

Specifically, considering a linear layer $F$ from UNet and its input $X \in \mathbb{R}^{n \times d}$ where $n$ is the number of token and $d$ is the feature dimension, we illustrate local and global assignment respectively.

**For local assignment**, we employ a local gating network that contains a learnable gating layer $G(\ , \phi)$ ($\phi \in \mathbb{R}^{d \times e}$, $e$ is the number of expert and here $e$ is 2.) and a $sigmoid$ function. The gating network is to produce two normalized score maps $s = [s_1, s_2] \in \mathbb{R}^{n \times 2}$ for each low-rank expert as formulated:

$$s = sigmoid(G(X\ , \phi) \tag{5}$$

**For global assignment**, we also use a gating network including an AdaptiveAvePool module, a learnable gating layer $G(\ , \omega)$ ($\omega \in \mathbb{R}^{d \times e}$, here $e$ is 2), and a $sigmoid$ function. This gating network is to produce two global scalars $g = [g_1,\ g_2] \in \mathbb{R}^2$ for each expert as formulated:

$$g = sigmoid(G(Pool(X)\ , \omega))\ . \tag{6}$$

*The soft mechanism is built on the fact that each token can adaptively determine how much (weight) should be sent to each expert by the $sigmoid$ function.* And intuitively, the weight of every token for two experts is independent as face and hand experts are not competitors in generation. Thus we do not use $softmax$.

**For combination**, we first send $X$ to each low-rank expert $E_{\text{face}}$ and $E_{\text{hand}}$ respectively, use $s_1$ and $s_2$ ($\mathbb{R}^{n \times 1}$) to perform element-wise multiplication (local assignment), and also perform global control by scalars $g_1$ and $g_2$ (global assignment) [7]:

$$Y_1 = E_{\text{face}}(X \cdot s_1 \cdot g_1) = g_1 \cdot E_{\text{face}}(X \cdot s_1) \quad Y_2 = E_{\text{hand}}(X \cdot s_2 \cdot g_2) = g_2 \cdot E_{\text{hand}}(X \cdot s_2) \tag{7}$$

Then we add $Y_1$ and $Y_2$ back to the output of a linear layer $F$ from UNet with $X$ as input, formulating a new output $X^{'}$:

$$X^{'} = F(X) + Y_1 + Y_2 \tag{8}$$

We use human-centric dataset to train the learnable parameters while freezing the base model and two low-rank experts.

## 4  EXPERIMENT

### 4.1  EVALUATION BENCHMARKS AND METRICS

Considering that our work primarily focuses on human-centric image generation, before presenting our experiment, we introduce two customized evaluation benchmarks. Additionally, since our generated images are human-centric, they should intuitively meet human preference. Hence, we adopt two human preference metrics including Human Preference Score (HPS) (Wu et al., 2023) and ImageReward (IR) (Xu et al., 2023). We describe all of them below. Besides the two metrics, we also perform user studies by inviting people to compare the generated images with their own preferences.

---

[7]Recalling that each expert is two low-rank matrixes, thereby $g_1$ and $g_2$ can transition from within $E_{\text{face}}$ and $E_{\text{hand}}$ to outside of them.

**Benchmark 1: COCO Human Prompts.** We construct this benchmark by leveraging the caption in COCO Caption (Chen et al., 2015) that has been widely used in previous work (Wu et al., 2023; Xu et al., 2023; Chen, 2023). Concretely, we use the captions in the COCO val set, and preserve the caption that contains human-related words, *e.g.*, woman, man, kid, girl, boy, person, teenager, *etc*. In the end, we have around 60k prompts left, dubbed as COCO Human Prompts.

**Benchmark 2: DiffusionDB Human Prompts.** We construct this benchmark by leveraging the caption in DiffusionDB 2M set (Wang et al., 2022) which is the first large-scale text-to-image prompt dataset. It contains 14 million images generated by Stable Diffusion using prompts specified by real users. Concretely, we first filter out the NSFW prompts by the indicator provided in DiffusionDB (Wang et al., 2022). Then we preserve captions containing human-related words. Additionally, we filter out prompts containing special symbol, *e.g.*, [, ], {, *etc*. In the end, we have around 64k prompts left, dubbed as DiffusionDB Human Prompts. We will release the two prompt sets for further research.

**Metric 1: Human Preference Score (HPS).** Human Preference Score (HPS) (Wu et al., 2023) measures how images present with human preference. It leverages a human preference classifier fine-tuned on CLIP (Radford et al., 2021).

**Metric 2: ImageReward (IR).** Different from HPS, ImageReward (IR) (Xu et al., 2023) is built on BLIP (Li et al., 2022) and is a zero-shot automatic evaluation metric for understanding human preference in text-to-image synthesis.

## 4.2 MAIN RESULTS

To evaluate the performance, following previous work (Rombach et al., 2022; Xu et al., 2023; Wu et al., 2023; Chen, 2023) we randomly sample 3k prompts from COCO Human Prompts benchmark and 3k prompts from DiffusionDB Human Prompts benchmark to generate images and calculate metrics, HPS and IR, and compare MoLE with open-resource SOTA method with different model structures including VQ-Diffusion (Gu et al., 2022), Versatile Diffusion (Xu et al., 2022), our baseline SD v1.5 (Rombach et al., 2022) and its largest variant SD XL. We repeat the process three times and report the averaged results and standard error as presented in Tab 1. MoLE outperforms VQ-Diffusion and Versatile Diffusion and significantly improves our baseline SD v1.5 in both metrics, implying that MoLE could generate images that are more natural to meet human preference. We also notice MoLE is inferior to SD XL in HPS and IR, as SD XL can generate images of high overall aesthetics. Due to that MoLE is built on SD v1.5, which has a significant gap in model size (5.1G *vs*. 26.4G) and output resolution (512 *vs*. 1024) compared to SD XL, it's difficult for MoLE to match with SD XL. Hence, we qualitatively compare the face and hand of generated images from MoLE and SD XL in Fig 8 and Fig 9, and our results are more realistic even under high HPS gap (in 2nd row Fig 8 20.39 *vs*. 22.38). Besides, MoLE is resource-friendly and can be trained in a single A100 40G GPU. Finally, we conduct user studies to further verify MoLE's advantage over baseline SD v1.5 by sampling 20 image pairs from both models and inviting 50 participants to select a better one from each pair according to their preference in four aspects respectively. We report the averaged results in Fig 5 where MoLE obtains higher voting, especially in hand and face quality.

## 4.3 ABLATION STUDY

### 4.3.1 STAGE ENHANCEMENT

Considering our MoLE contains three stages, figuring out how each stage enhances the generation performance is important. We conduct the experiments using COCO Human Prompts by randomly sampling 3k prompts to generate images from each stage with the same seed in Sec 4.2 and calculate the HPS and IR. The whole process repeats three times as well. The results are reported in Tab 2. It is observed that fine-tuning on human-centric dataset (Stage 1) is effective in remarkably improving the HPS and IR, implying the importance of the dataset. However, when adding Stage 2, *i.e.*, both experts are employed, the performance drops. We speculate the experts would influence the process of generation by resembling the distribution of the customized datasets due to training. We illustrate it in Fig 6. For example, the left image misses the "dog" and resembles the distribution of face image

Table 1: The performance of MoLE compared with SD v1.5 on COCO Human Prompts and DiffusionDB Human Prompts.

| Model | COCO Human Prompts | |
|---|---|---|
| | HPS (%) | IR (%) |
| VQ-Diffusion | $19.21 \pm 0.04$ | $-12.51 \pm 2.44$ |
| Versatile Diffusion | $19.75 \pm 0.09$ | $-8.81 \pm 1.40$ |
| SD XL | $20.84 \pm 0.06$ | $73.34 \pm 2.29$ |
| SD v1.5 | $19.91 \pm 0.09$ | $28.34 \pm 1.40$ |
| MoLE | $20.27 \pm 0.07$ | $33.75 \pm 1.49$ |

| Model | DiffusionDB Human Prompts | |
|---|---|---|
| | HPS (%) | IR (%) |
| VQ-Diffusion | $19.00 \pm 0.02$ | $-18.42 \pm 1.49$ |
| Versatile Diffusion | $20.09 \pm 0.04$ | $-29.05 \pm 2.72$ |
| SD XL | $21.51 \pm 0.07$ | $87.88 \pm 2.53$ |
| SD v1.5 | $20.29 \pm 0.01$ | $-2.72 \pm 1.66$ |
| MoLE | $20.62 \pm 0.04$ | $4.36 \pm 1.36$ |

Figure 5: User study in four aspects.

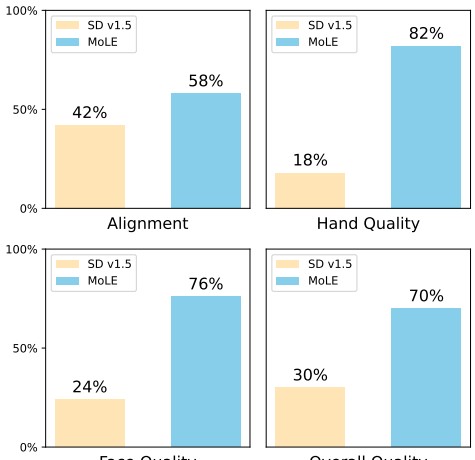

in FFHQ dataset (Karras et al., 2019). Adding Stage 3 alleviates this issue with mixture assignment and further enhances the performance.

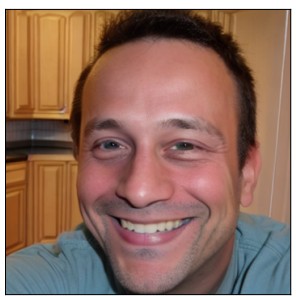 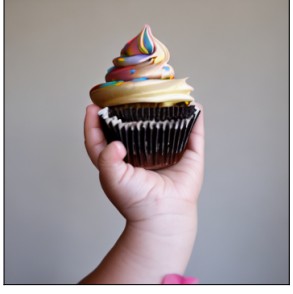

Figure 6: Showcases of disturbed data generation. Left prompt: a man in a gray jacket standing in a kitchen next to a black dog. Right prompt: a child holds a cupcake.

Table 2: Ablation study on each stage on COCO Human Prompts.

| Stage | HPS (%) | IR (%) |
|---|---|---|
| SD v1.5 | $19.91 \pm 0.09$ | $28.34 \pm 1.40$ |
| +Stage 1 | $20.16 \pm 0.09$ | $31.01 \pm 1.75$ |
| +Stage 2 | $19.94 \pm 0.07$ | $25.66 \pm 2.72$ |
| +Stage 3 | $20.27 \pm 0.07$ | $33.75 \pm 1.49$ |

Table 3: Ablation on different assignment on COCO Human Prompts.

| Method | HPS (%) | IR (%) |
|---|---|---|
| SD v1.5 | $19.91 \pm 0.09$ | $28.34 \pm 1.40$ |
| Local | $20.19 \pm 0.04$ | $31.98 \pm 2.41$ |
| Global | $20.20 \pm 0.02$ | $32.20 \pm 0.86$ |
| Both | $20.27 \pm 0.07$ | $33.75 \pm 1.49$ |

### 4.3.2 MIXTURE ASSIGNMENT

In MoLE, we use two kinds of mixture manners including local and global assignment. Hence, we ablate the two assignments and present the results in Tab 3. It can be seen that both local and global assignments can enhance performance. When combining them together, the performance is further improved, indicating the effectiveness of our method. Moreover, we present how the two assignments work in Fig 7. For the global assignment, we average the global scalars of 20 close-up face images, 20 close-up hand images, and 20 images involving hand and face respectively in every inference step in Fig 7 (a), (b), and (c). In (a) and (b), when generating different close-ups, the corresponding expert generally produces higher global value, implying that global assignment is content-aware. In (c), $E_{\text{face}}$ and $E_{\text{hand}}$ achieve a balance. Besides, as inference progresses the global scalar of $E_{\text{hand}}$ always drops while that of $E_{\text{face}}$ is relatively flat. We speculate, in light of the diversity of hands (*e.g.*, various gestures), $E_{\text{hand}}$ tends to establish general content in the early stage while $E_{\text{face}}$ must meticulously fulfill facial details throughout the denoise process due to fidelity requirement. For local assignment, we visualize the averaged score map of sampled images from different experts respectively in Fig 7(d). We see that as inference progresses, local assignment from different experts can highlight and gradually refine the corresponding parts.

### 4.4 VISUALIZATIONS

We present generated images and compare with other diffusion models as shown in Fig 8. It is observed that our method can generate more natural face and hand while aligning with given prompts.

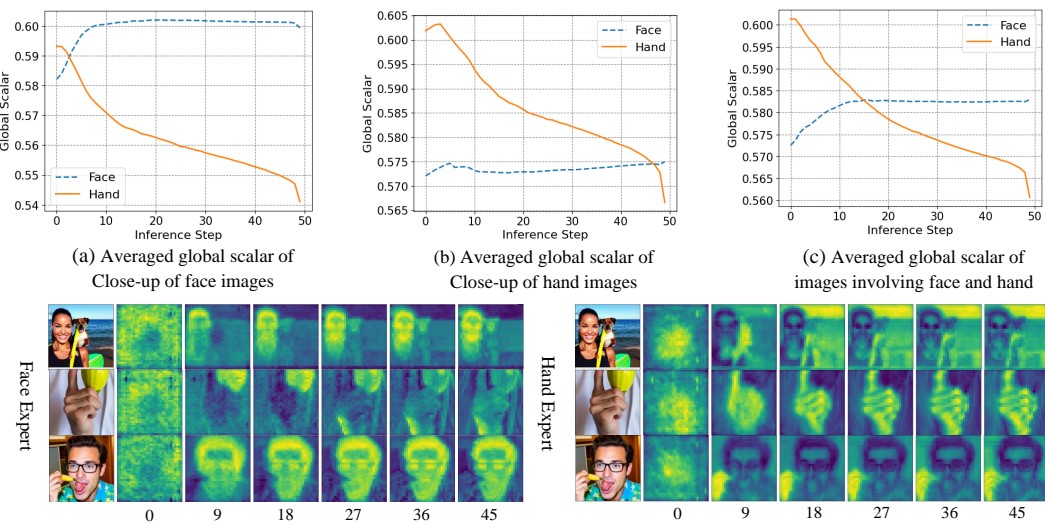

(a) Averaged global scalar of
Close-up of face images

(b) Averaged global scalar of
Close-up of hand images

(c) Averaged global scalar of
images involving face and hand

(d) Visualization of score map from face expert (left) and hand expert (right) in different inference step.

Figure 7: The averaged global and local assignment weights in different inference steps.

More results are put in Appendix A.4. We also find MoLE (fine-tuned SD v1.5) is capable of generating non-human-centric images, *e.g.*, animals, scenery, *etc*. [8]. See Appendix A.5.

# 5 RELATED WORK

**Text-to-image generation.** Diffusion models (Ho et al., 2020; Song et al., 2020) have been widely used in image generation since its proposal. Afterward, a vast effort has been devoted to exploring the applications, especially in text-to-image generation. GLIDE (Nichol et al., 2021) leverages two different kinds of guidance, *i.e.*, classifier-free guidance (Ho & Salimans, 2021) and clip guidance, to match the semantics of generated image with the given text. Imagen (Saharia et al., 2022) further improves the performance of text-to-image generation via a large T5 text encoder (Raffel et al., 2020). Stable Diffusion (Rombach et al., 2022) uses a VAE encoder to map image to latent space and perform diffusion on representation. DALL-E 2 (Ramesh et al., 2022) transfers text representation encoded by CLIP (Radford et al., 2021) to image representation via diffusion prior. Besides generation, diffusion model has also been used in text-driven image editing (Tumanyan et al., 2023). Inspired by the key observation between text and map in the cross-attention module, Prompt-to-prompt (Hertz et al., 2022) modifies the cross-attention map with prompt while preserving original structure and content. Further Null-text inversion (Mokady et al., 2023) achieves real image edition via image inversion. Different from these works, our work primarily focuses on human-centric text-to-image generation, aiming to alleviate the poor performance of diffusion model in this field.

**Mixture-of-Experts.** MoE is first proposed in (Jacobs et al., 1991). The underlying principle of MoE is that different subsets of data or contexts may be better modeled by distinct experts. Theoretically, MoE could scale model capability with little cost by using sparsely-gated MoE layer (Shazeer et al., 2017). (Lepikhin et al., 2020; Chen et al., 2023c;b) extend MoE on transformers by replacing FFN and attention layers with position-wise MoE layers. Swith Transformer (Fedus et al., 2022) simplifies the routing algorithm. (Zhou et al., 2022) propose Expert Choice (EC) routing algorithm to achieve optimal load balancing. GLaM (Du et al., 2022) scales transformer model parameter to 1.2T but is inference-efficient. VMoE (Riquelme et al., 2021) scale vision model to 15B parameter via MoE. Soft MoE (Puigcerver et al., 2023) introduces an implicit assignment by passing weighted combinations of all tokens to each expert. MoE has been adapted in generation to enhance the performance (Feng et al., 2023; Balaji et al., 2022; Jiang et al., 2022). For example, ERNIE-ViLG (Feng et al., 2023) uniformly divides the denoising process into several distinct stages, with each being associated with a specific model. eDiff-i (Balaji et al., 2022) calculates thresholds to separate the

---

[8]We speculate this is because the human-centric dataset also contains these entities that interact with humans in an image. As a result, the model learns these concepts. However, it is worth noting that our model may not be better at generic image generation than the generic models as MoLE is trained on human-centric images.

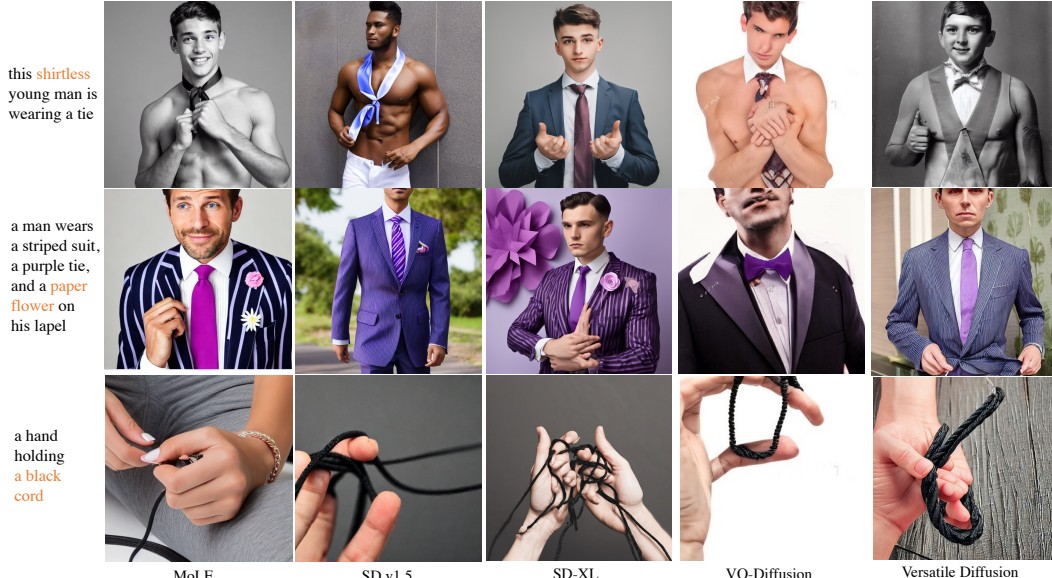

Figure 8: Comparison with other diffusion models. Zoom in for a better view.

whole process into three stages. Differing from employing experts in divided stages, we consider low-rank modules trained by customized datasets as experts to adaptively refine generation.

# 6    DISCUSSION

In this section, we present a comprehensive elaboration on the distinctions between MoLE and other mixture-of-experts approaches to further highlight our contribution.

There are three aspects of distinctions between MoLE and conventional mixture-of-experts approaches. Firstly, from the aspect of training, MoLE independently trains two experts with completely different knowledge using two customized close-up datasets. In contrast, conventional mixture-of-experts methods simultaneously train experts and base model using the same dataset.

Secondly, from the aspect of expert structure and assignment manner, MoLE simply uses two low-rank matrices while conventional mixture-of-experts methods use MLP or convolutional layers. Moreover, MoLE combines local and global assignments together for a finer-grained assignment while conventional mixture-of-experts methods only use global assignment.

Finally, from the aspect of applications in computer vision, MoLE is proposed for text-to-image generation while conventional mixture-of-experts methods are mainly used in object recognition, scene understanding, *etc.*, *e.g.*, V-MoE (Riquelme et al., 2021). Though MoE recently has been employed in image generation such as ERNIE-ViLG (Feng et al., 2023) and eDiff-i (Balaji et al., 2022) that employ experts in divided stages, MoLE differs from them and consider low-rank modules trained by customized datasets as experts to adaptively refine image generation.

# 7    CONCLUSION

In this work, we primarily focus on the human-centric text-to-image image generation that has important real-world applications but often suffers from producing unnatural results due to insufficient prior, especially the face and hand. To mitigate this issue, we carefully collect and process one million high-quality human-centric dataset, aiming to provide sufficient prior. Besides, we observe that a low-rank module trained on a customized dataset, *e.g.*, face, has the capability to refine the corresponding part. Inspired by it, we propose a simple yet effective method called Mixture of Low-rank Experts (MoLE) that effectively allows diffusion models to adaptively select experts to enhance the generation quality of corresponding parts. We also construct two customized human-centric benchmarks from COCO Caption and DiffusionDB to verify the superiority of MoLE.

**Limitation.** Our method may not be effective in a scenario involving multiple individuals. There could be two reasons. First, most of our collected images are single person. Secondly, there is un-

certainty regarding whether the observation in Fig 1 remains valid in a multiple individual scenario. More future work is expected in this direction.

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

# A   APPENDIX

## A.1   IMPLEMENTATION DETAILS

**Stage 1: Fine-tuning on human-centric Dataset.** We use Stable Diffusion v1.5 as base model and fine-tune the text encoder and UNet with a constant learning rate $2e - 6$. We set batch size to 64 and train with the Min-SNR weighting strategy (Hang et al., 2023). The clip skip is 1 and we train the model for 300k steps using Lion optimizer (Chen et al., 2023a).

**Stage 2: Low-rank Expert Generation.** For face expert, we set batch size to 64 and train it 30k steps with a constant learning rate $2e - 5$. The rank is set to 256 and AdamW optimizer is used. For hand expert, we set batch size to 64. Since ihans more sophisticated than face to generate, we train it 60k steps with a smaller learning rate $1e - 5$. The rank is also set to 256 and AdamW optimizer is used. For both experts, we only add low-rank module to UNet. And the two experts are both built on the fine-tuned base model in Stage 1.

**Stage 3: Mixture Adaptation.** In this stage, we use the batch size 64 and employ AdamW optimizer. We use a constant learning rate $1e - 5$ and train for 50k steps.

## A.2   LICENSE AND PRIVACY STATEMENT

The human-centric dataset is collected from websites including seeprettyface.com, unsplash.com, gratisography.com, morguefile.com, and pexels.com. We use web crawler to download images if it is allowed. Otherwise, we manually download the images. Most images in these websites are published by their respective authors under Public Domain CC0 1.0 [9] license that allows free use, redistribution, and adaptation for non-commercial purposes. Some of them require giving appropriate credit to the author, *e.g.*, seeprettyface.com requires adding the sentence (# Thanks to dataset provider:Copyright(c) 2018, seeprettyface.com, BUPT_GWY contributes the dataset.) to the open-source code when using the images. When collecting and filtering the data, we are careful to only include images that, to the best of our knowledge, are intended for free use and redistribution by their respective authors. That said, we are committed to protecting the privacy of individuals who do not wish their images to be included. Besides, for images fetched from other datasets, *e.g.*, Flickr-Faces-HQ (FFHQ) (Karras et al., 2019), Celeb-HQ (Karras et al., 2018), and 11k Hands (Afifi, 2019), we strictly follow their licenses and privacy.

## A.3   ILLUSTRATIONS OF NEGATIVE SAMPLES

In Fig 10, we present the illustrations of negative samples during refining human-in-the scene subset

## A.4   MORE VISUALIZATION

We present more generated images and compare with other diffusion models in Fig 9. Additionally, we also illustrate more images generated by MoLE in Fig 11, Fig 12, Fig 13, and Fig 14.

## A.5   GENERIC IMAGE GENERATION

As shown in Fig 15, MoLE ((fine-tuned SD v1.5)) can also generate non-human-centric images, *e.g.*, animals, scenery, *etc*. A main reason is that the human-centric dataset also contains these entities that interact with humans in an image. As a result, the model learns these concepts. However, intuitively, MoLE may not be better at generic image generation than the generic models as MoLE is trained on human-centric images.

---

[9]https://creativecommons.org/publicdomain/zero/1.0/

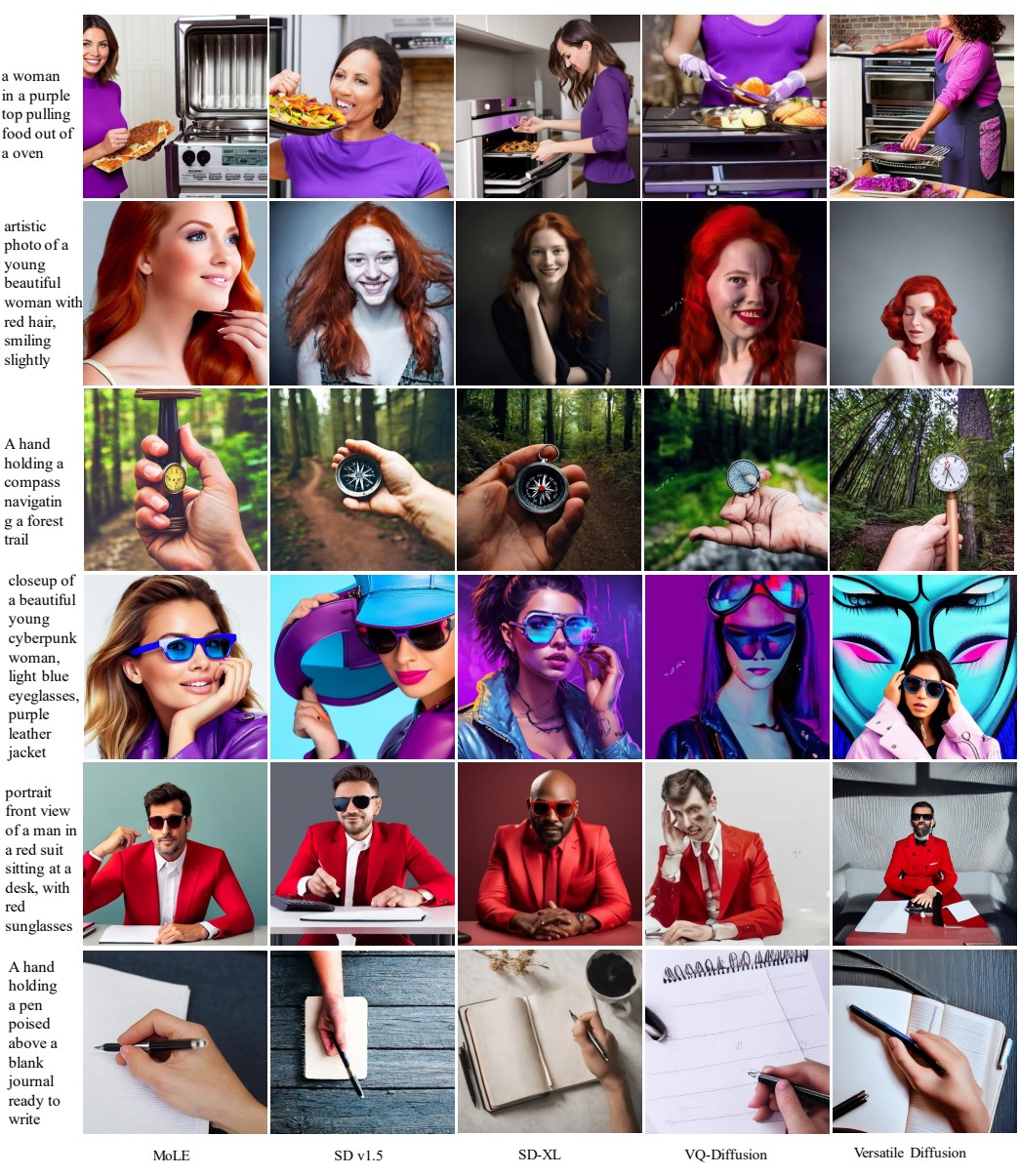

Figure 9: Comparison with other diffusion models. Zoom in for a better view.

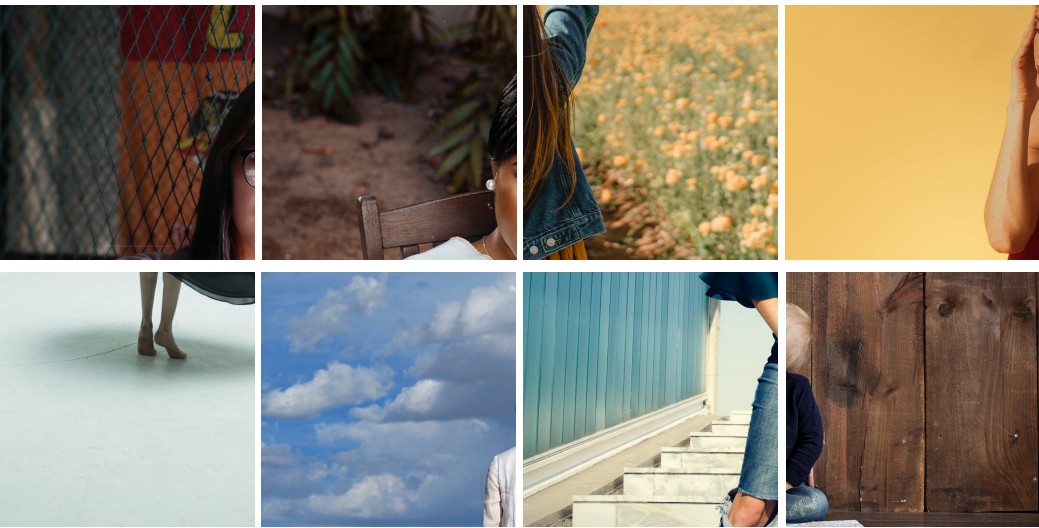

Figure 10: Illustrations of negative samples. Zoom in for a better view.

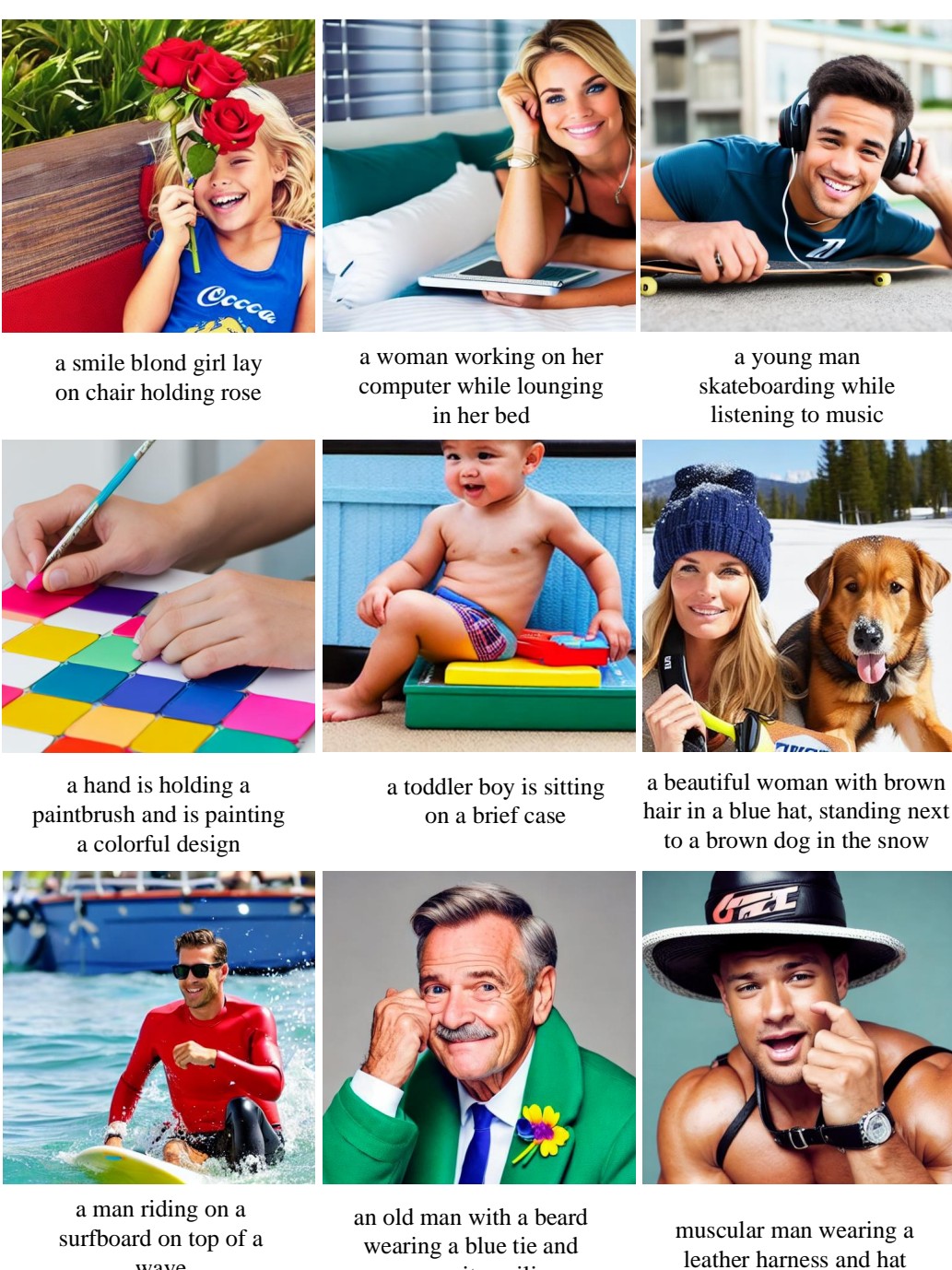

Figure 11: More images generated by MoLE. Zoom in for a better view.

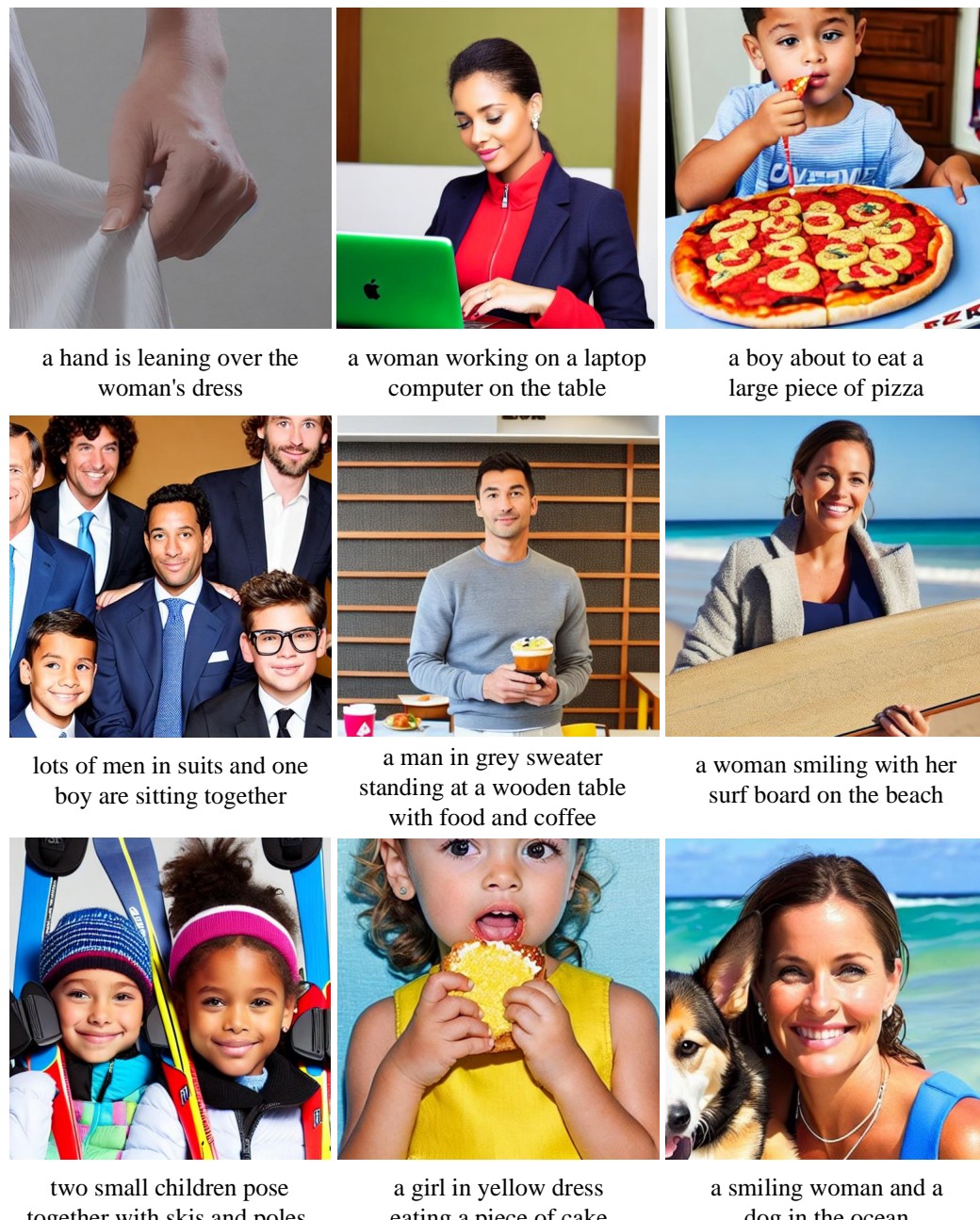

Figure 12: More images generated by MoLE. Zoom in for a better view.

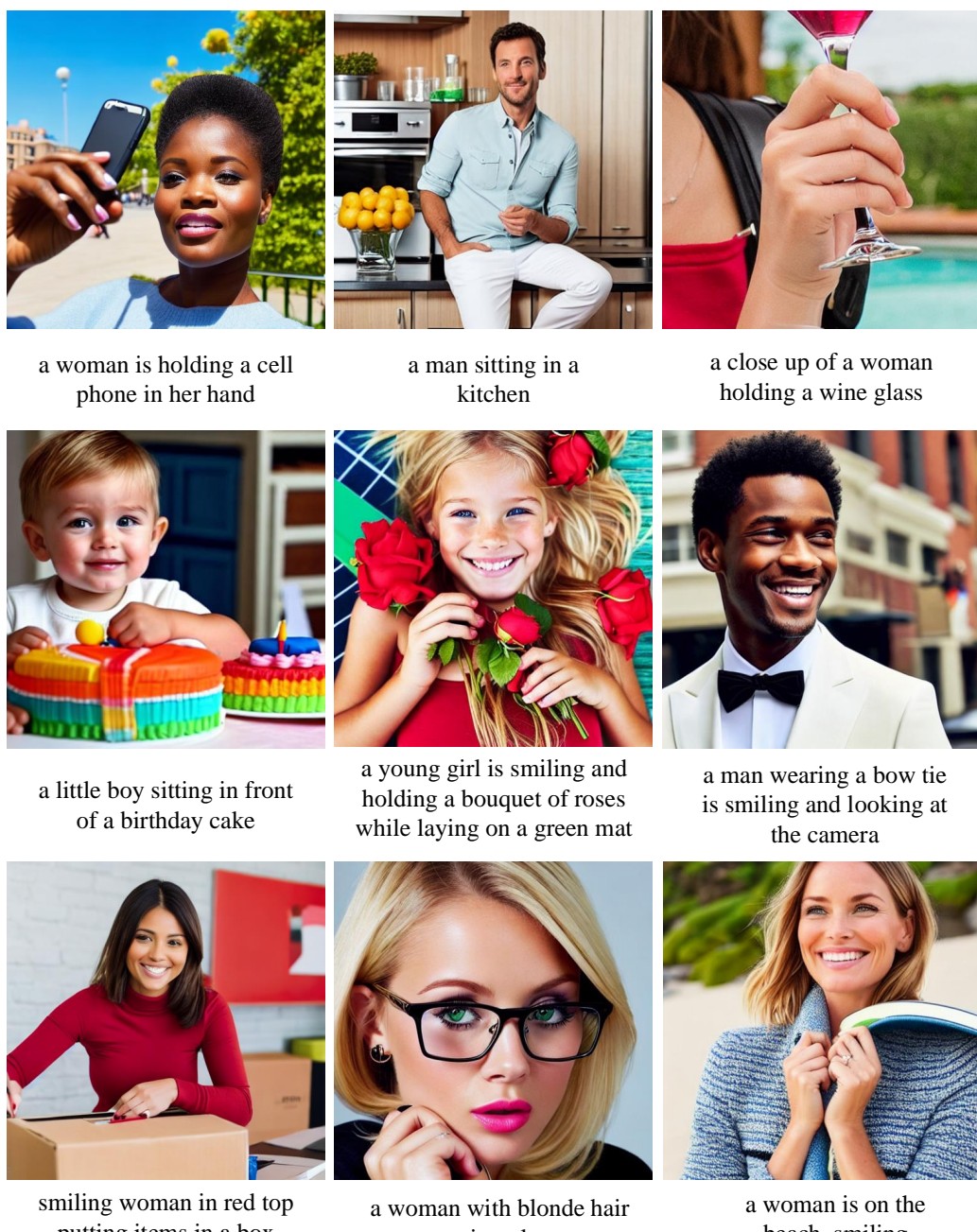

Figure 13: More images generated by MoLE. Zoom in for a better view.

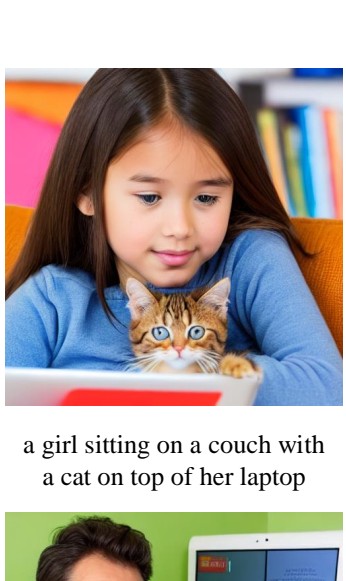
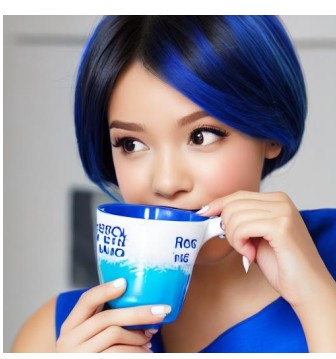
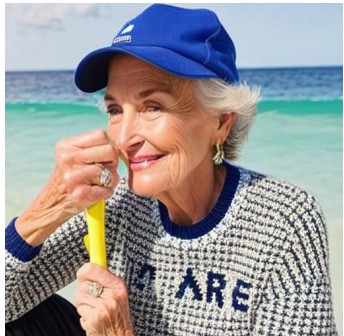

a girl sitting on a couch with a cat on top of her laptop

a woman with blue hair is drinking from a blue coffee cup

an older woman in a sweater sits at the beach

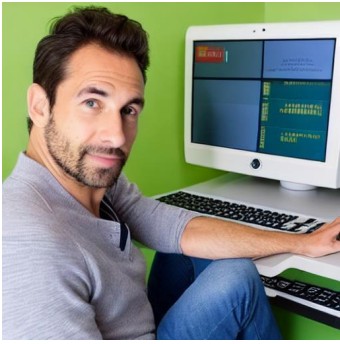
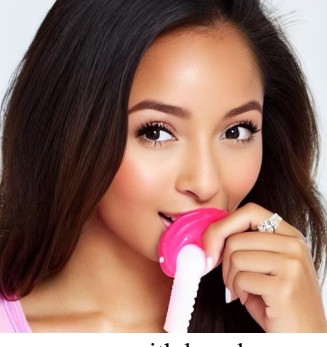
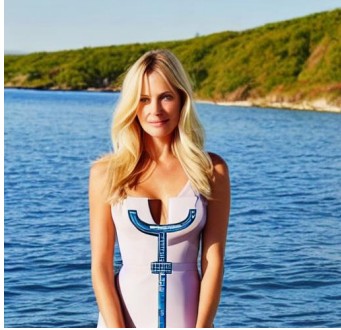

a man sitting in front of a large computer monitor

a woman with long brown hair is holding a pink toothbrush in her hand

a woman in a white dress stands on the edge of a lake

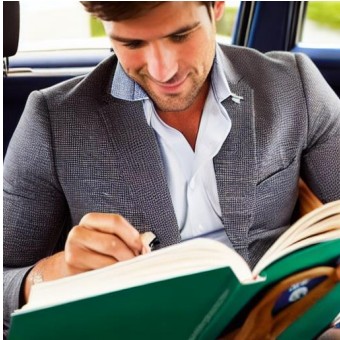
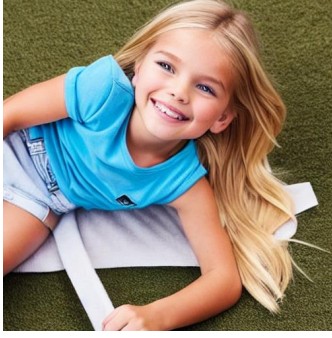
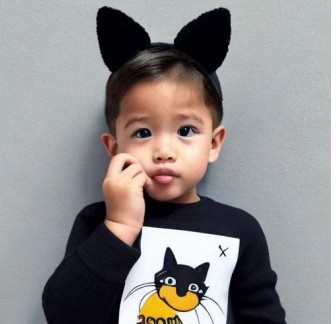

a man is sitting in a car reading a book

a pretty blonde haired girl wrapped up in a sheet and laying down

a young boy wearing a black shirt with a cat on it and cat ears

Figure 14: More images generated by MoLE. Zoom in for a better view.

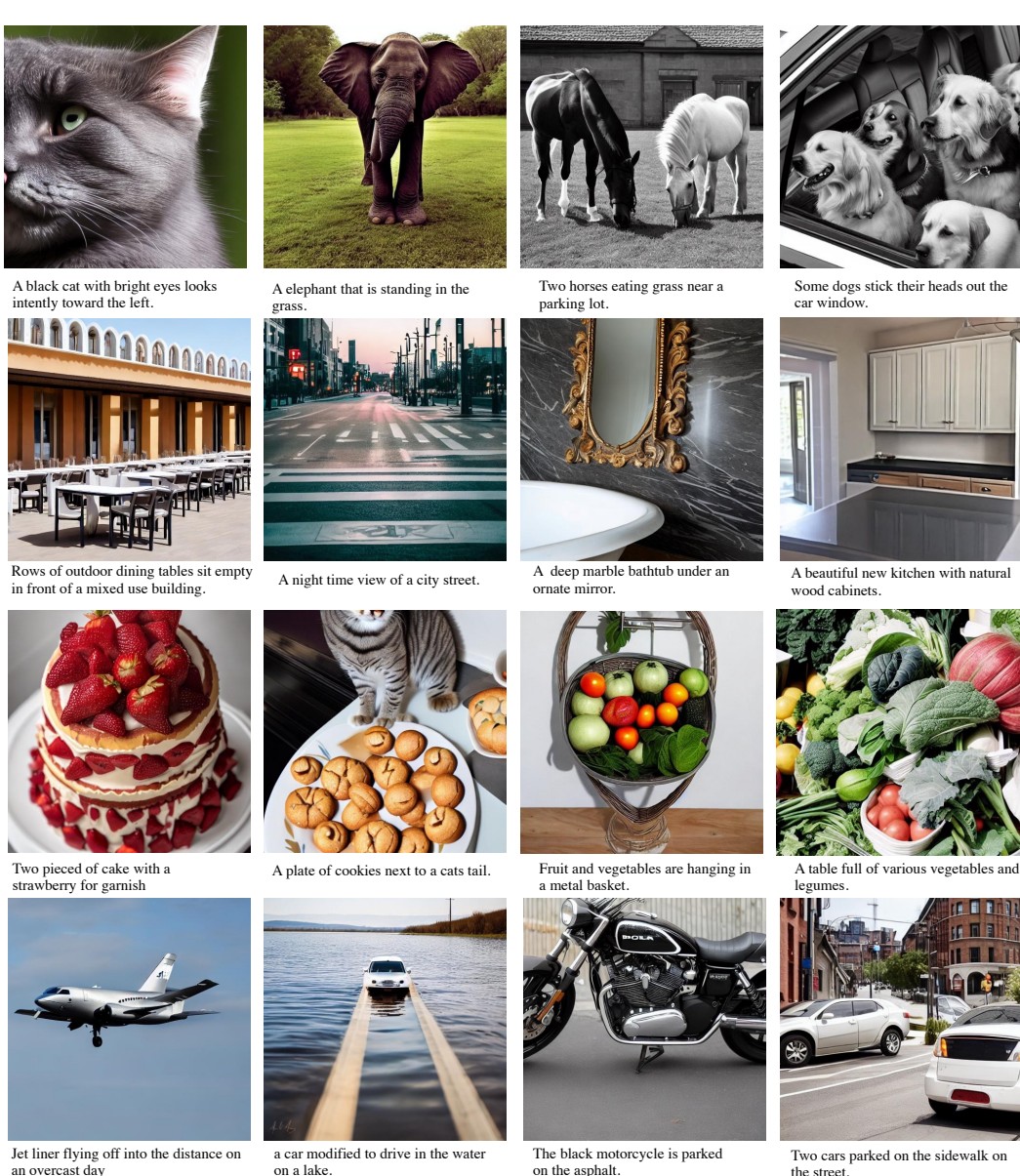

Figure 15: Generic image generation. Zoom in for a better view.

