# OpenReview forum: "MoLE: Human-centric Text-to-image Diffusion with Mixture of Low-rank Experts"
_ICLR.cc/2024/Conference — Submitted to ICLR 2024_

### Official Review · Reviewer_Y1yT · 2023-10-30

**Soundness:** 3 good
**Presentation:** 3 good
**Contribution:** 2 fair
**Rating:** 5
**Confidence:** 4

**Summary:**

This paper aims to solve the problem that existing text-to-image diffusion models sometimes fail to generate realistic human faces and hands. To this end, they collect several human face/hand specific datasets and use them to finetune the pretrained diffusion model. LoRA and MoE techniques are adopted for reducing updated-parameter scale and merging different trained modules respectively.

**Strengths:**

- This work moves a successful step towards generating more natural hands (especially) and faces.
- The proposed MoE strategy to allow different LoRA modules to work together is novel and makes sense.

**Weaknesses:**

It is obvious that recent progress in text-to-image generation, including this work, largely relies on the collected datasets. My main concern is about whether the collected datasets in this work would have ethical issues thus affect the model preference, given that it is a sensitive human-centric research topic, and unfortunately the collection and curation process is not comprehensively discussed. Since the "Human-in-the-scene images" and part of the "Close-up of hand images" are web-downloaded, they undoubtedly contain questionable contents. I am afraid I am not fully qualified to review this work from the ethical perspective and I recommend an ethics expert to assess this aspect. Nevertheless, I will try to list some concerns below:

- Is the web-collected dataset biased in terms of any subpopulations (e.g. race, age)?
- Do the generated image captions contain sensitive information of people (e.g. location, race, health situation, financial situation)?
- Are the raw data of original images logged (e.g. URL) for supporting community investigation and further improvement?

**Questions:**

- The scope of this work is somewhat over-claimed. "Human-centric" is a large scope where all human parts should be contained. But this work only considers the face and hand generation issue, leaving others like feet and leg not mentioned.

- The chosen SD v1.5 is somewhat an outdated baseline, since the stable diffusion team is also continually updating their model toward better generation of human content. In SD v2.1 (https://stability.ai/blog/stablediffusion2-1-release7-dec-2022, https://huggingface.co/stabilityai/stable-diffusion-2-1, released about 10 months ago), they claim that "the new release delivers improved anatomy and hands" by reducing the number of the filtered-out people in the dataset. It would be better to see whether the proposed method could still boost the human content generation quality upon this release.

- I have some concerns about reproducibility: whether the good results presented in the paper are carefully selected few ones. This can be addressed by
  - either providing the trained model and code (I haven't seen them on the provided anonymous webpage near the deadline of the review)
  - or conducting larger-scale quantitive experiments in Fig. 5 with SD XL or SD v2.1.

**Details Of Ethics Concerns:**

Please refer to the "weaknesses" part for my concerns.

---

> ### Author Response · Authors · 2023-11-18
> **Response to reviewer Y1yT**
>
> **Q1:** The scope of this work is somewhat over-claimed. "Human-centric" is a large scope where all human parts should be contained. But this work only considers the face and hand generation issue, leaving others like feet and leg not mentioned.
>
> **A1:** We thank the reviewer' comment. We may need to explain why we primarily focus on face and hand. There are two reasons:
>
> On the one hand, in Stage 1, due to the high quality and strong diversity of our human-centric dataset and its rich human-centric prior, we can generate better results for most of the human generation issues, e.g., poses, legs, arms, and overall body structure. The results of stage 1 in Table 2 show significant enhancement compared to SD v1.5. However, the human faces and hands are still relatively hard to generate naturally due to their high complexity and variability. Therefore,  to alleviate this issue, motivated by the low-rank refinement phenomenon shown in Figure 1, we design stages 2 and 3 of MoLE.
>
> Other the other hand, considering that the human face and hand are generally the most frequently observed parts in an image and their bad cases are also extensively discussed or complained in image generation communities, thereby we primarily focus on the two most important and urgent parts. Our work could also easily involve other human parts, e.g., feet, by collecting a close-up of the feet dataset, training an extra low-rank expert, and accordingly modifying the parameter of local and global assignment.
>
> We have added this discussion in our paper to avoid misunderstanding and overclaiming.
>
> **Q2:** The chosen SD v1.5 is somewhat an outdated baseline, since the stable diffusion team is also continually updating their model toward better generation of human content. In SD v2.1, they claim that "the new release delivers improved anatomy and hands" by reducing the number of the filtered-out people in the dataset. It would be better to see whether the proposed method could still boost the human content generation quality upon this release.
>
> **A2:** We thank the reviewer and follow this kind suggestion to perform an experiment on SD v2.1. We evaluate MoLE and SD v2.1 using COCO Human prompt. The experiment shows that MoLE produces $20.45 \pm 0.09$ for hps, outperforming SD v2.1 ($20.17 \pm 0.08$). This result further verifies the effectiveness of our methods.
>
> **Q3:** I have some concerns about reproducibility: whether the good results presented in the paper are carefully selected few ones.
>
> **A3:** For generated images presented in the paper, we repeat 3 times for each prompt and pick the best one for better visualization. And we also follow this process to generate images for other methods to ensure a fair comparison of visualizations in Figure 8 and Figure 9 and qualitatively verify the effectiveness of our method. Besides qualitative results, the quantitative results also demonstrate the effectiveness of our method compared to others. Additionally, we release our code and models [here](https://drive.google.com/drive/folders/1W8qKJG2jNQJ_Y2jlqn1w8VJu_c1ahLC2?usp=drive_link) for reproducibility.
>
> Finally, we feel sorry about this absence. Over the recent period, we concentrated on improving and expanding the size of our human-centric dataset (v2), e.g., merging more high-quality human-centric images by filtering other published datasets. More importantly, we have increased the amount of close-ups of hand images.  By manual cropping, we now have collected around **20k high-quality** close-ups of hand images. We present some of them at https://sites.google.com/view/response-close-up-of-hand/homepage . We are willing to release these high-quality close-ups of hand images to boost the development of image-generation communities.

---

> ### Author Response · Authors · 2023-11-22
>
> Dear Reviewer Y1yT：
>
> As the rebuttal period is ending soon, we wonder if our response answers your questions and addresses your concerns.
>
> Thanks again for your very constructive and insightful feedback!  Looking forward to your post-rebuttal reply!

---

> > ### Comment · Reviewer_Y1yT · 2023-11-22
> > **Thanks for your response**
> >
> > I thank the authors for providing details of the data collection process and other responses.
> > My main focus is whether our community could benefit from this work (and the collected dataset) in terms of boosting human content generation.
> > Given that the collected dataset is heavily biased in race, it is concerned about the trained model’s reliability for different human subpopulations.
> > And I tend to keep my score.
> >
> > P.S. I do not find the code or readme about how to generate images using the provided model from the provided code link. Nevertheless I trust the authors’ A3 response about the reproducibility and fair comparison to other methods.

---

> > > ### Author Response · Authors · 2023-11-22
> > > **Response to Reviewer Y1yT**
> > >
> > > We thank the reviewer for the timely reply.
> > >
> > > The reviewer asks whether the AIGC community could benefit from this work and the collected dataset in terms of boosting human content generation.
> > >
> > > We think it is a very important and serious question.  We try to answer this question in two folds including method and dataset.
> > >
> > > ## For method
> > >
> > > Firstly,  MoLE is motivated by the low-rank refinement phenomenon as shown in Figure 1 where a specialized low-rank module using **a proper scale weight** is capable of refining the corresponding part of person. **To the best of our knowledge, we are the first to discover and leverage this phenomenon.** And we believe this phenomenon would also inspire subsequent researchers to propose better methods.
> > >
> > > Secondly,  MoLE is an effective attempt to alleviate the issue of unnaturally generated human faces and hands, a problem of significant practical importance. We move a successful step towards generating more natural hands (especially) and faces. This would accelerate the application of human-centric image generation in reality.
> > >
> > > Finally, in terms of concrete implementation, our method proposes a new strategy to leverage the idea of MoE.  Previous generation methods leveraging MoE, e.g., ERNIE-ViLG and eDiff-i, primarily employ different experts to denoise the whole image in respective time stages.  Different from them, we primarily leverage a soft assignment mechanism in MoE to adaptively activate customized hand and face experts with suitable weights to refine the corresponding part.
> > >
> > > ## For dataset
> > >
> > > Our human-centric dataset has two aspects of advantages:
> > >
> > > - **Firstly, our human-centric dataset is of high quality.** As we mentioned in **A1 For Reviewer B5SW**, our human-centric dataset consists of images of high resolution (basically over 1024 × 2048) and large file size, e.g., 1M, 3M, 5M, 10M, etc collected from websites (e.g., [https://www.pexels.com/search/people/](https://www.pexels.com/search/people/), and [https://unsplash.com/s/photos/people](https://unsplash.com/s/photos/people)). The reviewer can simply click these links to have a see. In contrast, the average height and width of the human-centric images from LAION2b-en are approximately 455 and 415 respectively. Most of them are between 320 and 357. Compared to our human-centric dataset, they fail to provide sufficient and high-quality details.
> > > - **Secondly, our human-centric dataset is comprehensive.** As the face and hand are relatively hard to generate naturally compared to other parts as discussed in the AIGC community, besides the collected human-in-the-scene subset, we also provide two high-quality close-up of face and hand subsets, especially the close-up of hand images. **This is absent in human-centric images from LAION2b-en as mentioned in A1**. Additionally, through meticulous manual cropping, we now have curated 20k high-quality hand close-ups (https://sites.google.com/view/response-close-up-of-hand/homepage). **To the best of our knowledge, this quantity of close-up hand images is absent in prior related studies.** It will significantly help to address challenges associated with generating natural-looking hands and propel the advancement of human-centric image generation for subsequent researchers.
> > >
> > > These two advantages, we believe, have significant benefits to help generate high-quality human-centric images and accelerate its application in reality.
> > >
> > > Finally, the contribution of our proposed human-centric dataset is also acknowledged by **Reviewer FSG6 and 39sw**.
> > >
> > > As for the race bias, it is also common in other published human-centric datasets, e.g., face dataset (Celebrity in Places Dataset [1] (https://www.robots.ox.ac.uk/~vgg/data/celebrity_in_places) and FFHQ [2] (https://github.com/NVlabs/ffhq-dataset)). In Celebrity in Places Dataset, it shows the bias of celebrities. In FFHQ, it is said on its website that **the images were crawled from [Flickr](https://www.flickr.com/), thus inheriting all the biases of that website**. And it also has a large bias in country distribution which to some extent indicates the race bias.
> > >
> > > Similar to FFHQ, our human-centric dataset also inherits the potential bias from the target websites. We could alleviate this bias by incorporating more corresponding human-centric images from other published datasets.
> > >
> > > We thank the reviewer for proposing this important question and we will add this discussion in our paper to show our contribution to the community.  Looking forward to your reply.
> > >
> > > [1] Faces in Places: Compound Query Retrieval. BMVC 2016
> > >
> > > [2] A Style-Based Generator Architecture for Generative Adversarial Networks. CVPR 2019

---

> > > > ### Author Response · Authors · 2023-11-23
> > > >
> > > > Dear Reviewer Y1yT：
> > > >
> > > > We wonder if our response answers your questions and addresses your concerns.
> > > >
> > > > Thanks again for your previous reply! Looking forward to your another reply!

---

### Official Review · Reviewer_39sw · 2023-10-30

**Soundness:** 3 good
**Presentation:** 3 good
**Contribution:** 2 fair
**Rating:** 5
**Confidence:** 4

**Summary:**

Stable Diffusion Models (SDMs) often struggle to generate accurate facial and hand details, leading to unrealistic artefacts like ghost fingers. This paper attempts to tackle this problem and optimise SDMs for human-centric generation applications.
To achieve this, a dataset of one million human-focused images is collected, which includes three subsets of text-image pairs, close-up face images and close-up hand images.A multi-stage Mixture of Low-rank Experts training framework is proposed to leverage the different subsets and to adaptively combine the predictions from face and hand experts.
The results show that the proposed method empowers SDMs to create images with faithful facial and hand details. Moreover, the numerical metrics, measured by Human Performance Scores and ImageReward scores, surpass those of other text-image generation models, establishing a new state-of-the-art.

**Strengths:**

The dataset and benchmark contributions are valuable for developing data-hungry image-text generative models like diffusion models.
The paper is well-written and easy to follow.
The visualisations and accompanied website offer abundant visual examples.
Code, data, and benchmarks are promised to be open-sourced.

**Weaknesses:**

The proposed dataset has significant contributions, but lacks details and insights. Descriptions of the dataset distributions lack numerical support. For example, the authors claim "These images are diverse w.r.t. occasions, activities, gestures, ages, genders, and racial backgrounds" but there is no evidence to support this making the claim unsubstantiated.

The preprocessing of the human-in-the-scene dataset is vaguely explained. For example, they seem to "train a VGG19 to filter out images that contain little information about people" but how such images were identified and how the VGG19 was trained are not present.

Based on the description in Section 4.1 without diving into references, the classifiers used in evaluation metrics could inherit bias from training dataset. Such bias is especially harmful for human-centric applications. The bias is also present in visualisations where most generated humans have light skin colour.

The methodology section has limited contributions, with components being adaptations of existing works and lacking insightful studies of why the adapted components are necessary than other alternatives. It may be more appropriate to present these findings at a conference with a dedicated Dataset and Benchmark's track.

Lastly, the usefulness of the fine-tuned SDM in generic image generation is not addressed.

**Questions:**

* Is there any study on failure cases? For instance, in Figure 10, "a young man skateboarding while listening to music," an unrealistic half-man is generated, even though the hand and face appear natural. What role does each component play in causing such failures?
* As mentioned in the weaknesses, is there any quantitative evidence to support the diversity of the proposed dataset, apart from qualitative visualisations? For example, a gesture classifier and an object detector could be used to explore the variety within the hand dataset. A similar approach can be applied to other subsets as well.
* What does "adding stage 2" mean in section 4.3.1? There are two low-rank experts trained on two separate close-up datasets (one for faces and the other for hands). Are both experts utilised in the "stage 2" ablation, and if so, how are their predictions combined? It would be beneficial to compare this approach with a simpler mixture method, such as directly adding the two predictions.
* As there is already large amount of AI-generated images online, how was it guaranteed that all the collected images are real during the web-crawling stage?

**Details Of Ethics Concerns:**

- The constructed human-centric dataset may contain unfair distributions and no fairness study has been conducted.
- The authors do not have consents for distributing the images that contain human faces. This may not comply with GDPR or other local laws and the images could be misused by companies if published.

---

> ### Author Response · Authors · 2023-11-18
> **Response to Reviewer 39sw (1/2)**
>
> **Q1:** The proposed dataset has significant contributions, but lacks details and insights. Descriptions of the dataset distributions lack numerical support. For example, the authors claim "These images are diverse w.r.t. occasions, activities, gestures, ages, genders, and racial backgrounds" but there is no evidence to support this making the claim unsubstantiated.
>
> **A1:** We thank the reviewer for pointing our this issue. We use [deepface](https://github.com/serengil/deepface) to  identify the race distribution where approximately 57.33% of individuals identify as white, 14.68% as Asian, 9.98% as black, 5.11% as Indian,  5.52% as Latino Hispanic,  and 7.38% as Middle Eastern. We also find approximately 58.18% are male and 41.82% are female. We also use an age classifier to identify the age distribution where approximately 0.93% are babies (0-1 years old), 3.55% are kids (2-11 years old), 4.60% are teenagers (12-18 years old), 84.86% are adults (18-60 years old), and 6.06% are elderly (over 60 years old). We have added these evidences in our paper to support our statement and sincerely thank the reviewer for this constructive suggestion.
>
> **Q2:** The preprocessing of the human-in-the-scene dataset is vaguely explained. For example, they seem to "train a VGG19 to filter out images that contain little information about people" but how such images were identified and how the VGG19 was trained are not present.
>
> **A2:** We are sorry for the confusion. In the preprocessing, we regard an image where the most area is covered by the background as a negative sample except those that major body is approximately centered in the image. We give several intuitive cases of the negative sample in  (See Visualization for Q2 in https://sites.google.com/view/response-reviewer-39sw/homepage). Images that do not meet this standard are regarded as positive samples. However, to be honest, considering that the collection criterion is relatively vague, could not be quantified, and thereby varies among different people, the collected samples inevitably have a certain degree of subjective bias. To alleviate this issue, we try our best to keep justice and repeat the filtering process multiple times as we mentioned in our paper.
>
> To train the VGG19, we manually collect around 300 positive samples and 300 negative samples as training set, and we also collect around 100 positive samples and 100 negative samples as val set. When training the VGG19, we set batch size to 128, set learning rate to 0.001, and use random flip as data augmentation method. We train the model for 200 epochs and use the best-performing model for subsequent classification. We have added the training details and illustrations of negtive samples in our paper.
>
> **Q3:** Lastly, the usefulness of the fine-tuned SDM in generic image generation is not addressed.
>
> **A3:** We thank the reviewer for this kind suggestion. Our model also can generate non-human-centric images. To verify it, we try some prompts without human-related keywords  (See Visualization for Q3 in https://sites.google.com/view/response-reviewer-39sw/homepage). Results show that our method is capable of generating high-quality and diverse images including non-human-centric entities, e.g., animals, scenery, etc. We speculate this is because the human-centric dataset also contains these entities that interact with humans in an image. As a result, the model learns these concepts. However, it is worth noting that our model may not be better at generic image generation than the generic models as MoLE is trained on human-centric images. We have added the discussion to our paper.
>
> **Q4:** Is there any study on failure cases? For instance, in Figure 10, "a young man skateboarding while listening to music," an unrealistic half-man is generated, even though the hand and face appear natural. What role does each component play in causing such failures?
>
> **A4:** We thank the reviewer for this constructive suggetion. To figure out the reason, we sample 10 bad and normal cases, calculate their L2 norm of outputs from face and hand expert respectively, and visualize the averaged L2 norm across timestep. (See Visualization for Q4 in https://sites.google.com/view/response-reviewer-39sw/homepage) One can see that the bad cases generally have larger L2 norm for both experts, which indicates that the output from linear layer $F$ in Eq (8) is strongly influenced by the two experts. As a result, the generated images may be uncoordinated.

---

> ### Author Response · Authors · 2023-11-18
> **Response to Reviewer 39sw (2/2)**
>
> **Q5:** What does "adding stage 2" mean in section 4.3.1? There are two low-rank experts trained on two separate close-up datasets (one for faces and the other for hands). Are both experts utilised in the "stage 2" ablation, and if so, how are their predictions combined? It would be beneficial to compare this approach with a simpler mixture method, such as directly adding the two predictions.
>
> **A5:** Yes, both experts are utilized in stage 2 ablation. We rephrase the sentence in our paper for clarity as follows:
>
> However, when adding Stage 2, i.e., both experts are employed, the performance drops.
>
> As for the second question, we may need to explain that in stage 2 ablation, the two experts are both combined with base model together at the same time. Recalling that both experts (low-rank matrices) are employed on the **linear layer (a matrix)**, thereby the whole process is a linear calculation, which means their predictions are directly added together. As for the results, they are exactly illustrated in Figure 6.
>
> **Q6:** As there is already large amount of AI-generated images online, how was it guaranteed that all the collected images are real during the web-crawling stage?
>
> **A6:** Our human-centric dataset is collected from websites including unsplash.com, gratisography.com, seeprettyface.com, morguefile.com, and pexels.com. These websites are established relatively early. For example, according to Wikipedia, unsplash.com is established in 2013 and Pexels.com is established in 2014. This would ensure that most of our collected images are real. Hence, even if there are some images generated by AI, their proportion is very small. To further decrease the proportion, we can use a fake detector to exclude images that may potentially be generated by AI.

---

> ### Author Response · Authors · 2023-11-22
>
> Dear Reviewer 39sw：
>
> As the rebuttal period is ending soon, we wonder if our response answers your questions and addresses your concerns.
>
> Thanks again for your very constructive and insightful feedback!  Looking forward to your post-rebuttal reply!

---

> > ### Author Response · Authors · 2023-11-23
> >
> > Dear Reviewer 39sw：
> >
> > We wonder if our response answers your questions and addresses your concerns.
> >
> > Thanks again!  Sincerely looking forward to your reply!
> >
> > the authors

---

### Official Review · Reviewer_FSG6 · 2023-10-31

**Soundness:** 2 fair
**Presentation:** 2 fair
**Contribution:** 2 fair
**Rating:** 6
**Confidence:** 3

**Summary:**

In the human-centric text-to-image generation, particularly in the context of faces and hands, the results often lack naturalness due to insufficient training priors. The authors address this problem from two perspectives. Firstly, on the data front, they create a human-centric dataset comprising approximately one million high-quality person-scene images, along with two distinct sets of close-up facial and hand images. Secondly, in terms of methodology, they propose MoLE, which involves incorporating low-rank modules trained separately on close-up hand and facial images as experts.

**Strengths:**

1. This paper focuses on addressing the issue of stable diffusion models having subpar generation results for human faces and hands, a problem of significant practical importance.
2. The high-quality human-centric dataset constructed in this work will effectively advances research in this area.
3. As shown in Figure 7, Mixture-of-experts make low-rank modules work together well. The assembly of two separate modules, each excelling in distinct functionalities, results in a natural and effective approach.

**Weaknesses:**

1. The method has obvious limitations, as it requires the collection of substantial amounts of data. How to measure the quality of such data, what about the computational burden of using such data, etc, remain unresolved issues. Besides, the method is ineffective in scenarios involving multiple individuals would also limit its practical value.
2. Many methods employed in this paper are already in existence, with the primary innovation residing in the "mixture of low-rank experts" (MoLE). However, the paper lacks a comprehensive elaboration on the distinctions between MoLE and other mixture-of-experts approaches.
3. I am curious on the results in Table 1. The numbers are hard to evaluate on the performance improvement of this work.

**Questions:**

Please check the weakness.

---

> ### Author Response · Authors · 2023-11-18
> **Response to Reviewer FSG6  (1/2)**
>
> **Q1:** The method has obvious limitations, as it requires the collection of substantial amounts of data. How to measure the quality of such data, what about the computational burden of using such data, etc, remain unresolved issues. Besides, the method is ineffective in scenarios involving multiple individuals would also limit its practical value.
>
> **A1:** We may need to emphasize that collecting such a large amount (around 1M) of high-quality images from websites is also one of our contributions. It took us around one month to finish. And to advance the development of the research community, we are willing to release this high-quality dataset, which would significantly save the time and effort of subsequent researchers.
>
> In terms of quality, most of our data is obtained from well-known free websites that contain exquisite high-resolution images, e.g., https://www.pexels.com/search/people/, and https://unsplash.com/s/photos/people. The quality is relatively high.
>
> Regarding the computational load, subsequent researchers can leverage a portion of our dataset to strike a balance between effectiveness and computational burden after the dataset is released. It is worth mentioning that MoLE is resource-friendly and can be trained in a single A100 40G GPU.
>
> Finally, the limitation in scenarios involving multiple individuals is primarily because most of our collected images are single person. Since we have verified the effectiveness of MoLE in scenarios involving a single individual,  we can strategically expand the dataset to involve more images with multiple individuals to mitigate this issue, which we leave as future work.
>
> **Q2:** Many methods employed in this paper are already in existence, with the primary innovation residing in the "mixture of low-rank experts" (MoLE). However, the paper lacks a comprehensive elaboration on the distinctions between MoLE and other mixture-of-experts approaches.
>
> **A2:** We thank the reviewer for the constructive suggestion that would further highlight our contribution and difference from other mixture-of-experts approaches. Below is a comprehensive elaboration on the distinctions between MoLE and other mixture-of-experts approaches:
>
> There are three aspects of distinctions between MoLE and conventional mixture-of-experts approaches. Firstly, from the aspect of training, MoLE independently trains two experts with completely different knowledge using two customized close-up datasets. In contrast, conventional mixture-of-experts methods simultaneously train experts and base model using the same dataset.
>
> Secondly, from the aspect of expert structure and assignment manner, MoLE simply uses two low-rank matrices while conventional mixture-of-experts methods use MLP or convolutional layers. Moreover, MoLE combines local and global assignments together for a finer-grained assignment while conventional mixture-of-experts methods only use global assignment.
>
> Finally, from the aspect of applications in computer vision, MoLE is proposed for text-to-image generation while conventional mixture-of-experts methods are mainly used in object recognition, scene understanding, etc., e.g., V-MoE[1]. Though MoE recently has been employed in image generation such as ERNIE-ViLG[2] and eDiff-i[3] that employ experts in divided stages, MoLE differs from them and consider low-rank modules trained by customized datasets as experts to adaptively refine image generation.
>
> We have created a new section for this discussion. Please see Section Discussion. We are also willing to cite more related work if the reviewer provides.

---

> ### Author Response · Authors · 2023-11-18
> **Response to Reviewer FSG6 (2/2)**
>
> **Q3:** I am curious on the results in Table 1. The numbers are hard to evaluate on the performance improvement of this work.
>
> **A3:** Since reviewer is curious about the results in Table 1, we describe how we obtain such scores in detail. By doing so, we hope to prove that the score is obtained justly and is effective in validating the performance improvements. Specifically, we randomly sample 3k prompts from two human benchmarks to generate images and calculate metrics, HPS and IR, using their released official evaluation model. Afterward, we average the produced HPS and IR scores for the 3k prompts. We repeat the whole process three times and use the averaged HPS and IR to calculate the mean and standard deviations. Besides, To further verify the effectiveness, we also adopt another two metrics including clip alignment and FID to evaluate the performance. We follow BLIP-Diffusion[4] and HumanSD[5] to report the results using COCO Human Prompts and Human-centric datasets. Clip alignment is to measure the text-image alignment (the higher, the better) and FID is to measure the distribution distance between generated images and real images (the lower, the better). MoLE produces 27.33 for clip alignment and outperform SD v1.5 (26.87). MoLE produces 64.37 for FID and outperform SD v1.5 (69.82). These results indicate that MoLE is also superior to SD v1.5.
>
> [1] Scaling Vision with Sparse Mixture of Experts NeurIPS2021
>
> [2] Ernie-vilg 2.0: Improving text-to-image diffusion model with knowledge-enhanced mixture-of-denoising-experts. CVPR2023
>
> [3] ediffi: Text-to-image diffusion models with an ensemble of expert denoisers. Arxiv2023
>
> [4] BLIP-Diffusion: Pre-trained Subject Representation for Controllable Text-to-Image Generation and Editing. Arxiv2023
>
> [5] HumanSD: A Native Skeleton-Guided Diffusion Model for Human Image Generation. ICCV2023

---

> ### Author Response · Authors · 2023-11-22
>
> Dear Reviewer FSG6：
>
> As the rebuttal period is ending soon, we wonder if our response answers your questions and addresses your concerns.
>
> Thanks again for your very constructive and insightful feedback!  Looking forward to your post-rebuttal reply!

---

### Official Review · Reviewer_B5SW · 2023-11-02

**Soundness:** 2 fair
**Presentation:** 1 poor
**Contribution:** 2 fair
**Rating:** 5
**Confidence:** 5

**Summary:**

Since an open-sourced text-to-image (T2I) model, Stable Diffusion v1.5, has a limitation in generating human-centric images, this study has tried to alleviate this problem in terms of data and model aspects. The authors carefully collect human-centric images, including publicly available datasets such as CelebA and FFHQ, and then fine-tune Stable Diffusion v1.5 on the collected dataset. In addition, this study proposes a Mixture of Low-rank Experts to further fine-tune the T2I model, while a method of soft MoE is adopted especially for increasing the generation quality of faces and hands. On the proposed benchmark, the fine-tuning improves the quality of generated images with respect to human preferences such as HPS and IR scores.

**Strengths:**

S1. This study aims to resolve a well-known limitation of open-sourced T2I models, which lead to low-quality of human-centric images including faces and hands.

S2. The experimental results show that fine-tuning on human-centric datasets can improve the quality of generated images in terms of human preference.

S3. The main idea, global and Local assignments of two LoRA experts, is novel and makes sense, while having potential to be further improved and expanded.

**Weaknesses:**

W1. The detailed information and statistics of collected data are missing. To ensure the quality of collected human-centric images, the authors need to analyze the characteristics and attributes of the dataset in a systematic and quantitative manner. For example, the distribution of racial groups and gender, the diversity of text prompts including semantic context, or the distribution of face size and the number of faces have to be provided to understand the characteristics of the dataset.

W2. The organization of presentation of this paper should be improved. This paper includes some types and grammatical errors. In addition, the detailed explanation of the proposed method, MoLE, is not provided, while presenting only the overall idea. For example, I cannot find the detailed explanation about how the LoRA experts and gating model $G$ are formulated.

W3. Although the design of the proposed method, MoLE, makes sense, its design is not well-motivated. Please refer to the questions below for the details.

W4. The experimental results are limited to demonstrate the effectiveness of the proposed idea, while the experiments can be considered unfair. Please refer to the questions below for the details.

W5. This paper does not include a discussion about negative social impacts.

**Questions:**

Q1. In Section 1, the authors postulate “the absence of comprehensive and high-quality human centric data within the training dataset LAION5B…”, but the claim does not have supporting analysis. Is there any specific analysis to show that LAION5B does not include comprehensive and high-quality human centric images?

Q2. In Section 1, the authors also claim “faces and hands represent the two most complicated parts due to high variability, making them challenging to be generated naturally.” However, although MoLE adopts a soft mixture of LoRAs, the capability of T2I model is not much increased compared with Stable Diffusion v1.5. Can the proposed method synthesize naturally generated faces and hands, while significantly outperforming previous methods? Figure 1 and Figure 8 still show that the proposed method provides unnaturally generated eyes and hands.

Q3. In my opinion, one of the main reasons why a T2I model cannot generate high-quality faces and hands is that complex attributes and details of faces and hands are often located in a small region, considering synthesizing high-quality images with small objects is difficult for T2I models. I wonder the authors’ opinion and the reason why the authors focus on close-up faces and hands for fine-tuning in Stage 2 and Stage 3.

Q4. Can the proposed MoE exploit numerous LoRA experts more than two? Can the proposed method support the adoption of multiple LoRA experts for face and hand, respectively?

Q5. What are the details of each low-rank expert $E_\text{face}$ and $E_\text{hand}$? Since they are not formally defined and described, I wonder the exact operation (including mathematical formulations) of each expert. In addition, the operation of the learnable gating layer $G$ is not formally defined.

Q6. My major question is whether the comparison of experimental results is fair or not. First, since MoLE further trains SD v1.5, the results that outperforms SD v1.5 are natural and do not support the effectiveness of MoLE. In addition, I wonder why the CLIP text encoder is further trained together with U-Net. How about the user study to compare MoLE and fine-tuned SD v1.5?

Q7. After fine-tuning of SD v1.5 in Stage 1, is the model capable of generating high quality and diverse images including non-human-centric entities such as animals, foods, landscapes, and etc?

Q8. Why is the LoRA applied to the U-Net blocks? What if the LoRA is applied to the (key, value) projection layers and MLP layers in self- and cross-attention blocks of SD?

Q9. The authors claim that the degradation of performance in Stage 2 results from overfitting. That is, does the training loss decrease but validation loss increase?  In addition, can the model after Stable 1 successfully generate images for the prompts in Figure 6?

Q10. This paper describes that 3K prompts are used in Section 4.2 and 1K prompts are used in Section 4.3.2. However, why are the reported performances (mean, std)  of MoLE in Table 1 and Table 2 (+Stage 3) exactly the same? Since Table 2 uses random sampling of 1K prompts among the 3K prompts, the HPS and IR scores cannot be exactly the same with the reported values in Table 1, considering the standard deviations (0.07, 1.49).

Q11. The authors first use local assignments and then use global assignments in Figure 4. Does the ordering of the two affect the experimental results?

Q12. Instead of using Stage 2 and Stage 3, which use 30K+60K and 50K training steps, respectively, how about the results of Stage 1 when we train the model longer than 300K steps?

Q13. There are minor questions about the experiments.
- The authors should also report CLIP similarity, FID, and aesthetic score to evaluate the overall quality of generated images in addition to human preferences.
- In Table 1, how the standard deviations are measured?
- Can the proposed approach also improve the performance of SD-XL?
- I wonder why the authors do not present generated images on user-provided text prompts that rarely appear in the training and benchmark prompts.
- In Figure 7(d), why do the face and hand experts also highlight non-human-centric regions?

---

> ### Author Response · Authors · 2023-11-18
> **Response to Reviewer B5SW (1/5)**
>
> We are happy that Reviewer B5SW is quite interested in our work and raises many interesting questions. We carefully consider each question the reviewer proposed and sincerely hope that our pointwise response below could remove the concerns. If it is achieved or the reviewer have new question, do not hesitate to let us know.
>
> **Q1:** In Section 1, the authors postulate “the absence of comprehensive and high-quality human centric data within the training dataset LAION5B…”, but the claim does not have supporting analysis. Is there any specific analysis to show that LAION5B does not include comprehensive and high-quality human centric images?
>
> **A1:** Yes, there are two reasons. Firstly, in our preliminary experiments, we analyzed human-centric images from LAION2b-en. We randomly sample 35w human-centric images and find that the average height and width are 455 and 415 respectively. Most of them are between 320 and 357. Compared to the image we collected (basically over 1024 × 2048), the image in LAION2b-en fails to provide sufficient and high-quality details.
>
> Secondly, in human-centric images from LAION2b-en, We randomly sample 1000 images each time and repeat the process three times, in which we find little (almost zero) high-quality close-ups of face and hand (especially hand). This makes the human-centric images from LAION2b-en relatively limited in providing comprehensive human-centric knowledge. In contrast, based on such motivation, our work bridges this gap by proposing a relatively comprehensive and high-quality human-centric dataset. We will add these specific analysis to our paper to support our opinion.
>
> **Q2:** In Section 1, the authors also claim “faces and hands represent the two most complicated parts due to high variability, making them challenging to be generated naturally.” However, although MoLE adopts a soft mixture of LoRAs, the capability of T2I model is not much increased compared with Stable Diffusion v1.5. Can the proposed method synthesize naturally generated faces and hands, while significantly outperforming previous methods? Figure 1 and Figure 8 still show that the proposed method provides unnaturally generated eyes and hands.
>
> **A2: We do not agree with the reviewer that the capability of MoLE is not much increased compared with Stable Diffusion v1.5.** We take metric HPS on COCO for example. As shown in Table 1, compared with SD v1.5, SD-XL improves HPS by 0.93% and increases the model size from 4.3G to 26.4G. In contrast, MoLE improves HPS by 0.36% but only increases the model size by 0.8G, which means MoLE is more efficient. The results also indicate that MoLE is resource-friendly. MoLE can be trained in a single A100 40G GPU.
>
> Moreover, in Figure 8 and Figure 9, by qualitatively comparing the face and hand of generated images from MoLE and SD XL, we find that SD XL also generates unnatural hand and unrealistic face while our results are more realistic. When it comes to SD v1.5 in Figure 8 and Figure 9, the generated results are even worse. Our user study in Figure 5 also supports our conclusions, especially the hand quality, face quality, and overall quality.
>
> Finally, we have to point out that Figure 1 is not the result of our method. It is just an illustration of our motivation.

---

> ### Author Response · Authors · 2023-11-18
> **Response to Reviewer B5SW (2/5)**
>
> **Q3:** In my opinion, one of the main reasons why a T2I model cannot generate high-quality faces and hands is that complex attributes and details of faces and hands are often located in a small region, considering synthesizing high-quality images with small objects is difficult for T2I models. I wonder the authors’ opinion and the reason why the authors focus on close-up faces and hands for fine-tuning in Stage 2 and Stage 3.
>
> **A3:** We agree with the reviewer's opinion that complex attributes and details of faces and hands are often located in a small region. **That is also one of the reasons why we collect the close-up of face and hand besides the human-in-the-scene subset.** Moreover, we use two customized low-rank experts to learn specific knowledge from the two close-up datasets. It is worth mentioning that we are still scaling the data quantity. By manual cropping, we now have collected around 20k high-quality close-ups of hand images.
>
> We also want to supplement another two reasons. Firstly, as we mentioned in our paper, humans have abundant face expressions, (e.g., laughing and crying) and hand gestures (spread and curve), resulting in high variability which is also considerably challenge for generation model. Secondly, the hand and face are the most familiar and sensitive subjects for human observation, If there's even a slight anomaly, humans can readily perceive it. Hence, from this view, the standard of generating natural face and hand is strict, which also makes face and hand difficult to synthesize.
>
> As for the reason, we are motivated by the low-rank refinement phenomenon in Figure 1. We observe that the low-rank module is capable of refining the corresponding part when using a proper scale weight. Hence, in Stage 2, we use the two close-up datasets to train two low-rank modules as experts. In Stage 3, we introduce local and global assignments in a MoE form. By training, we hope them to adaptively activate different experts with suitable weights.
>
> **Q4:** Can the proposed MoE exploit numerous LoRA experts more than two? Can the proposed method support the adoption of multiple LoRA experts for face and hand, respectively?
>
> **A4:** Yes is the answer.  It is easy for MoLE to incorporate more LoRA experts. Specifically, we can train numerous LoRA experts in stage 2 and in stage 3 adjust the parameter e in local assignment gating layer in Eq (5) and parameter value 2 in global assignment gating layer in Eq (6) to the corresponding number of involved experts. Considering that we primarily focus on alleviating the unnatural results of face and hand, we adopt two experts, i.e., face and hand experts. Regarding the second question, it parallels the first one, involving multiple experts concentrating on the same aspect, such as the face. Our method is also compatible. For example, we can train three face experts for the white race, black race, and yellow race respectively. Also, we can train two face experts for women and men respectively. Then follow the modification described in the first question.
>
> **Q5:** What are the details of each low-rank expert Eface and Ehand? Since they are not formally defined and described, I wonder the exact operation (including mathematical formulations) of each expert. In addition, the operation of the learnable gating layer G is not formally defined.
>
> **A5:** We speculate that the reviewer may be confused by the illustration of Figure 4 "Fine-tuned UNet layer" in stage 3. We deeply apologize and feel a profound sense of remorse for this confusion as for the sake of aesthetics,  we overlooked some details. We may need to raise the reviewer's attention to the presented caption in Figure 4 and the description in Eq (8) and in Section 3.1 Low-rank Adaptation (LoRA) part. In MoLE, for **every linear layer of the UNet**, each expert consists of two low-rank matrixes. As for the learnable gating layer $G$, **it is a linear layer parameterized by $\phi$ and $\omega$ in local and global assignment respectively**. We will redraw Figure 4 and emphasize the operation position to avoid potential confusion. We also thank the reviewer for proposing the question to catch our attention.

---

> ### Author Response · Authors · 2023-11-18
> **Response to Reviewer B5SW (3/5)**
>
> **Q6:** My major question is whether the comparison of experimental results is fair or not. First, since MoLE further trains SD v1.5, the results that outperforms SD v1.5 are natural and do not support the effectiveness of MoLE. In addition, I wonder why the CLIP text encoder is further trained together with U-Net. How about the user study to compare MoLE and fine-tuned SD v1.5?
>
> **A6:** We understand the reviewer's concern. However, we are not conducting continuous training on the dataset utilized for training SD v1.5; rather, we train it on a new dataset with a distinct distribution, namely, our human-centric dataset. More importantly, comparing the proposed method with baseline is quite common in previous works [1][2][3][4][5][6] that also leverage new datasets and propose new methods. We believe that this comparison is to verify the effectiveness of the whole strategy including collected dataset and proposed algorithm (note that not only the algorithm). Consequently, to verify the effectiveness of each, in our work, we compare the enhancements of stage 1 and stage 3 that correspond to the effectiveness of collected dataset and proposed algorithm. Please see Section 4.3.1 and Tab.2. In Tab 2, stage 3 further improves the performance, verifying the effectiveness of proposed algorithm.
>
> As for the reason of jointly training CLIP text encoder and UNet, we simply follow BLIP-Diffusion[6] and this is verified in BLIP-Diffusion to be helpful for better image-text alignment. See Table 2 in BLIP-Diffusion.
>
> Finally, we follow the reviewer's advice and conduct a user study to compare with fine-tuned SD v1.5 following the process reported in our paper. The results below show that MoLE still obtains higher voting compared to fine-tuned SD v1.5, especially in hand and face quality.
>
> | User Study | Alignment | Hand Quality | Face Quality | Overall Quality |
> | -------- | -------- | -------- |-------- | -------- |
> | Fine-tuned SD v1.5     | 46%     | 20%    |28%     | 34%     |
> | MoLE     | 54%    |80%    | 72%    |66%     |
>
> **Q7:** After fine-tuning of SD v1.5 in Stage 1, is the model capable of generating high quality and diverse images including non-human-centric entities such as animals, foods, landscapes, and etc?
>
> **A7:** It is an interesting question. We try some prompts without human-related key words from COCO Caption. Results shows that it is capable to generate high quality and diverse images including non-human-centric entities. (See Visualization for Q7 in https://sites.google.com/view/response-reviewer-b5sw/homepage). We speculate this is because that the human-centric dataset also contains these entities that interact with human in an image. As a result, the model learn these concepts. But we may need to acknowledge that it may not be superior over SD v1.5 in the aspect of non-human-centric generation as our dataset primarily involves human-centric images.
>
> **Q8:** Why is the LoRA applied to the U-Net blocks? What if the LoRA is applied to the (key, value) projection layers and MLP layers in self- and cross-attention blocks of SD?
>
> **A8:** Please see A5.
>
> **Q9:** The authors claim that the degradation of performance in Stage 2 results from overfitting. That is, does the training loss decrease but validation loss increase? In addition, can the model after Stable 1 successfully generate images for the prompts in Figure 6?
>
> **A9:** We do not observe such a phenomenon. Here, we mean that the generated image resembles the distribution of the training dataset. For example, in our experiment, face expert is trained on the close-up of face called FFHQ where a human's face covers the most area of an image. Thus, when equipped with face expert, the generated image could be similar to the training data like Figure 6 (left) in which the face also covers most area of the image. We will rephrase the sentence for clarity.
>
> Secondly, for the reviewer's second question, we try to generate image and find the model after Stage 1 can successfully generate images for the prompts in Figure 6, But the details of face and hand are not so good.  (See Visualization for Q9 in https://sites.google.com/view/response-reviewer-b5sw/homepage)
>
> **Q10:** This paper describes that 3K prompts are used in Section 4.2 and 1K prompts are used in Section 4.3.2. However, why are the reported performances (mean, std) of MoLE in Table 1 and Table 2 (+Stage 3) exactly the same? Since Table 2 uses random sampling of 1K prompts among the 3K prompts, the HPS and IR scores cannot be exactly the same with the reported values in Table 1, considering the standard deviations (0.07, 1.49).
>
> **A10:** Sorry for the confusion. It is a writing mistake by accident. We also use 3k prompts in Table 2.

---

> ### Author Response · Authors · 2023-11-18
> **Response to Reviewer B5SW (4/5)**
>
> **Q11:** The authors first use local assignments and then use global assignments in Figure 4. Does the ordering of the two affect the experimental results?
>
> **A11:** We are sorry for the confusion. In fact, the two assignments are performed at the same time. See the first equation in $Y_1$ and $Y_2$ in Eq (7). We may have to emphasize that the global assignment produces a **global scalar** for each expert and recall that each expert is two low-rank matrixes. Thus, $g_1$ and $g_2$ can transition from within the function $E$ to outside of it. We will add this discussion to our paper for clarity.
>
> **Q12:** Instead of using Stage 2 and Stage 3, which use 30K+60K and 50K training steps, respectively, how about the results of Stage 1 when we train the model longer than 300K steps?
>
> **A12:** We may need to emphasize our core purpose, i.e., **improving the generation of the hands and face**, and our stages 2 and 3 are necessary for these two specific purposes. If we only preserve Stage 1, i.e., fine-tuning the model on the human-centric dataset longer, in general, we believe the model would perform better overall than fine-tuned SD v1.5 trained by 300K step, e.g., in image-text alignment and overall quality, if not overfitting. However, considering the proportion of close-ups of hand and face images in the whole human-centric dataset, around 0.7%, like a long-tail problem, it is hard for the model to balance its capability,  and thereby the model could not learn sufficient knowledge about hand and face. So it could also not generate natural hand and face. Intuitively, using two extra customized experts to learn specific knowledge seems more reasonable. The user study in A6 also supports our speculation as the hand quality and face quality are still inferior to that of our MoLE.
>
> Moreover, recall that we train the text encoder and UNet together in Stage 1, this would consume a large amount of computation resources. In contrast, in Stage 2 we freeze the base model and only train the low-rank experts. In Stage 3 we freeze the base model and the low-rank experts and only train local and global gating layers. The whole process is more resource-friendly as less parameters are learnable.
>
> **Q13:** The authors should also report CLIP similarity, FID, and aesthetic score to evaluate the overall quality of generated images in addition to human preferences.
>
> **A13:** We thank the reviewer's advice. We follow [4] and [6] using COCO Human Prompts and Human-centric datasets to evaluate the overall quality of generated images on metrics including the CLIP similarity, and FID as reported below. We also use an aesthetic predictor (https://laion.ai/blog/laion-aesthetics/) to generate the aesthetic score.
>
> | Model | CLIP sim | FID | Aesthetic Score |
> | -------- | -------- | -------- | -------- |
> | SD v1.5     | 26.87     | 69.82     | 5.19     |
> | MoLE     | 27.33     | 64.37    | 5.18     |
>
> We find MoLE is also superior in CLIP similarity and FID. We also find that MoLE is slightly inferior to SD v1.5 in aesthetic score. We deem that it is reasonable as SD v1.5 is especially fine-tuned on **laion-aesthetics v2 5+ dataset** in which **each image's aesthetic score is evaluated with high aesthetics score by exactly the aesthetic predictor we used in this comparison**.
>
> **Q14:** In Table 1, how the standard deviations are measured?
>
> **A14:** As we describe in Section 4.2,  we randomly sample 3k prompts from two human benchmarks to generate images and calculate metrics, HPS and IR, using their released official evaluation model. Afterward, we average the produced HPS and IR scores for the 3k prompts.  We repeat the whole process three times and use the averaged HPS and IR to calculate the mean and standard deviations.
>
> **Q15:** Can the proposed approach also improve the performance of SD-XL?
>
> **A15:** Yes is the answer. As shown in Figure 8 and Figure 9, we also see that SD-XL fails to generate correct, natural, and realistic face and hand (especially hand) while our method can be employed on any linear layer. Therefore, it is also feasible to apply MoLE on SD-XL. We are trying to actively reach out to members of stability.ai, hoping them to adopt our method.

---

> ### Author Response · Authors · 2023-11-18
> **Response to Reviewer B5SW (5/5)**
>
> **Q16:**  I wonder why the authors do not present generated images on user-provided text prompts that rarely appear in the training and benchmark prompts.
>
> **A16:** We may need to explain that the benchmark, i.e., DiffusionDB, is exactly collected from real users. On its website (https://github.com/poloclub/diffusiondb) it is reported that **DiffusionDB is the first large-scale text-to-image prompt dataset. It contains 14 million images generated by Stable Diffusion using prompts specified by real users.** That is also one of the reasons we select DiffusionDB as our benchmark, aiming to evaluate how our method performs under users' prompts in reality. Hence, we actually indeed present generated images on user-provided text prompts, e.g., "portrait front view of a man in a red suit sitting at a desk, with red sunglasses", and "artistic photo of a young beautiful woman with red hair, smiling slightly" in Figure 9, "muscular man wearing a leather harness and hat" and "a smile blond girl lay on chair holding rose" in Figure 10, etc.
>
> **Q17:** In Figure 7(d), why do the face and hand experts also highlight non-human-centric regions?
>
> **A17:** It is easy to explain. When the target, i.e., face or hand does not appear in the image (e.g., in the second row of Figure 7(d) left, the face is absent), the corresponding expert may be disturbed by noise feature in the generation process.
> Luckily, due to the existence of global assignment, this interference can be alleviated because global assignment could produce a relatively small scalar as shown in Figure 7(a) and (b) to gradually reduce such interference.
>
> [1]  BLIP: Bootstrapping Language-Image Pre-training for Unified Vision-Language Understanding and Generation. ICML2022
>
> [2] Visual Instruction Tuning. NeurIPS2023
>
> [3] Generative Pretraining in Multimodality. Arxiv2023
>
> [4] HumanSD: A Native Skeleton-Guided Diffusion Model for Human Image Generation. ICCV2023
>
> [5] RAPHAEL: Text-to-Image Generation via Large Mixture of Diffusion Paths. Arxiv2023
>
> [6] BLIP-Diffusion: Pre-trained Subject Representation for Controllable Text-to-Image Generation and Editing. Arxiv2023

---

> ### Author Response · Authors · 2023-11-22
>
> Dear Reviewer B5SW：
>
> As the rebuttal period is ending soon, we wonder if our response answers your questions and addresses your concerns.
>
> Thanks again for your very constructive and insightful feedback!  Looking forward to your post-rebuttal reply!

---

> > ### Comment · Reviewer_B5SW · 2023-11-22
> >
> > Thanks for the detailed authors' responses, and very sorry for the late reply.
> >
> > After I read the authors' responses, I still have some concerns about the contributions of proposed dataset and the novelty of the proposed method, since fine-tuning LoRA weights for a specific purpose is a widespread approach.
> > However, I agree that this paper has a strong motivation to improve human-centric image generation.
> >
> > I will consider the authors' responses and the strong points, during the discussion periods with other reviewers.

---

> ### Author Response · Authors · 2023-11-22
> **Response to Reviewer B5SW**
>
> We sincerely thank the reviewer for the timely reply.
>
> As for the reviewer's concern about the contribution of proposed dataset and the novelty of MoLE, we give a detailed answer below:
>
> ## For the contribution of proposed dataset
>
> Our human-centric dataset has two aspects of advantages:
>
> - **Firstly, our human-centric dataset is of high quality.** As we mentioned in **A1**, our human-centric dataset consists of images of high resolution (basically over 1024 × 2048) with large file size, e.g., 1M, 3M, 5M, 10M, etc, collected from websites (e.g., [https://www.pexels.com/search/people/](https://www.pexels.com/search/people/), and [https://unsplash.com/s/photos/people](https://unsplash.com/s/photos/people)). The reviewer can simply click these links to have a see. In contrast, the average height and width of the human-centric images from LAION2b-en are approximately 455 and 415 respectively. Most of them are between 320 and 357. Compared to our human-centric dataset, they fail to provide sufficient and high-quality details.
> - **Secondly, our human-centric dataset is comprehensive.** As the face and hand are relatively hard to generate naturally compared to other parts as discussed in the AIGC community, besides the collected human-in-the-scene subset, we also provide two high-quality close-up of face and hand subsets, especially the close-up of hand images. **This is absent in human-centric images from LAION2b-en as mentioned in A1**. Additionally, through meticulous manual cropping, we now have curated 20k high-quality hand close-ups (https://sites.google.com/view/response-close-up-of-hand/homepage). **To the best of our knowledge, this quantity of close-up hand images is absent in prior related studies.** It will significantly help to address challenges associated with generating natural-looking hands and propel the advancement of human-centric image generation for subsequent researchers.
>
>   Finally, the contribution of our proposed human-centric dataset is also acknowledged by **Reviewer FSG6 and 39sw**.
>
> ## For the novelty of MoLE
>
> The proposed method, MoLE, is primarily motivated by the low-rank refinement phenomenon as shown in Figure 1 where a specialized low-rank module using **a proper scale weight** is capable of refining the corresponding part of person.
>
> We agree with the reviewer that fine-tuning LoRA weights for a specific purpose is a widespread approach. **However, it is only one stage of our method, i.e., Stage 2, to prepare the customized experts.**
>
> In fact, inspired by the low-rank refinement phenomenon, **the novelty of MoLE lies in Stage 3 where we leverage a soft assignment mechanism in MoE to adaptively activate different experts with suitable weights.** Additionally, we introduce an extra **element-wise local assignment** and combine it with a normal global assignment to achieve finer-grained activation. The effectiveness is also verified by the ablation results in Table 3.
>
> Our method is also different from other generation methods leveraging MoE, e.g., ERNIE-ViLG and eDiff-i. They employ different experts to denoise the whole image in respective stages. In contrast, we primarily leverage a soft assignment mechanism in MoE to adaptively activate customized hand and face experts to refine the corresponding part.
>
> Finally, the novelty of our method is also acknowledged by **Reviewer Y1yT**.
>
> We hope our answer would further remove your concern and looking forward to your reply.

---

> > ### Author Response · Authors · 2023-11-23
> >
> > Dear Reviewer B5SW：
> >
> > We wonder if our response answers your questions and addresses your concerns.
> >
> > Thanks again for your previous reply! Looking forward to your another reply!

---

### Author Response · Authors · 2023-11-18
**Response to Ethics Concern**

We sincerely appreciate all reviewers' time and effort in reviewing our paper.  We also thank reviewers for considering the ethical and legal implications of our work. Below is our response to the potential ethics concern to reviewers.

**For collected image:**
**Fairness:** We conduct fairness assessments encompassing three dimensions: race, age, and gender to identify potential biases. Regarding race, approximately 57.33% of individuals identify as white, 14.68% as Asian, 9.98% as black, 5.11% as Indian,  5.52% as Latino Hispanic,  and 7.38% as Middle Eastern. Regarding gender, approximately 58.18% are male and 41.82% are female. Regarding age, approximately 0.93% are babies (0-1 years old), 3.55% are kids (2-11 years old), 4.60% are teenagers (12-18 years old), 84.86% are adults (18-60 years old), and 6.06% are elderly (over 60 years old).

**Legal Compliance:** The human-centric dataset is collected from websites including unsplash.com, gratisography.com, seeprettyface.com, morguefile.com, pexels.com, etc. Images on these websites are published by their respective authors under Public Domain CC0 1.0 3 license that allows free use, redistribution, and adaptation for non-commercial purposes. When collecting and filtering the data, we are careful to only include images that, to the best of our knowledge, are intended for free use. We are committed to protecting the privacy of individuals who do not wish their images to be included. Besides, for images fetched from other datasets, e.g., Flickr-FacesHQ (FFHQ) [1], Celeb-HQ [2], and 11k Hands [3], we strictly follow their licenses and privacy requirements, e.g., we only use them for our academic research. Finally, when releasing this dataset, we promise to emphasize its exclusive use for academic purposes only, and carefully select a suitable license to ensure compliance with legal regulations.

**For generated caption:**
The caption is generated by LLAVA which is trained on COCO image-text pairs in a multimodal instruction-following formulation  constructed by ChatGPT/GPT-4. To the best of our knowledge, COCO and ChatGPT/GPT-4 have adopted some privacy protection strategies to avoid illegal information and are relatively safe. For example, in ChatGPT, information like name, address, and phone number are anonymized. Hence, it may not be possible to generate detailed information, e.g., private location, health situation, financial situation, etc., about a specific person. We also checked the generated caption and did not find proper nouns related to diseases, private places, name, etc., to specific person.

We also thank all reviewers for their insightful and constructive suggestions that help a lot in further improving our paper. According to reviewer's suggestions, we have revised our paper. Below are the main modifications (highlighted in blue):

- Add a discussion to clarify the concentration of our work to avoid misunderstanding and overclaiming
- Add fairness assessments encompassing three dimensions: race, age, and gender to identify potential biases, and provide more details about our human-centric dataset.
- Give a detailed description of how we train VGG19 to preprocess the dataset
- Discuss generic image generation issue and illustrate some results
- Add more discussion about the discrepancy between MoLE and other conventional mixture-of-experts methods to highlight the contribution of our work.
- Rephrase sentences, fix typo issues, and polish the figure.

Finally, we thank all reviewer's effort again and hope our pointwise responses below could clarify each reviewers' confusion.

[1]  A style-based generator architecture for generative adversarial networks. CVPR 2019

[2] Progressive growing of gans for improved quality, stability, and variation. ICLR 2018

[3]  11k hands: Gender recognition and biometric identification using a large dataset of hand images. MTA 2019

---

### Author Response · Authors · 2023-11-20

Dear Reviewer B5SW, FSG6, 39sw, Y1yT, and AC:

We pen down this letter with a nervous disposition as the discussion is going to be close.

We sincerely hope that the reviewers and AC will take into consideration the contributions of our method and (more preciously) the recently gathered dataset comprising one million high-quality human-centric images, especially the close-ups of hand.  Through meticulous manual cropping, we have curated 20k high-quality hand close-ups (https://sites.google.com/view/response-close-up-of-hand/homepage ). We are willing to release such close-ups as we strongly believe that it will greatly help to address challenges associated with generating natural-looking hands and propel the advancement of human-centric image generation.

We are also very grateful to the reviewers for their constructive comments and suggestions and hope to obtain feedback from reviewers.

The authors

Best

---

### Author Response · Authors · 2023-11-23
**General Response about Race Bias and Contribution Summary**

## Response for race bias

Considering that one of the reviewers is concerned about race bias, we deem that we may need to give an explanation to remove such concern from the reviewer and potentially other reviewers.

For the race bias, it is also common in other published human-centric datasets, e.g., face dataset (Celebrity in Places Dataset [1] (https://www.robots.ox.ac.uk/~vgg/data/celebrity_in_places) and FFHQ [2] (https://github.com/NVlabs/ffhq-dataset)):

In Celebrity in Places Dataset, it shows the bias of celebrities.

 In FFHQ, it is said on its website that **the images were crawled from [Flickr](https://www.flickr.com/), thus inheriting all the biases of that website**. And it also has a large bias in country distribution which to some extent indicates the race bias.

Similar to FFHQ, our human-centric dataset also inherits the potential bias from the target websites. We could alleviate this bias by incorporating more corresponding human-centric images from other published datasets.

## Contribution summary

Our contribution lies in two folds including method and dataset:

### For method

- **Firstly,  MoLE is an effective attempt to alleviate the issue of unnaturally generated human faces and hands, a problem of significant practical importance.** We move a successful step towards generating more natural hands (especially) and faces. This would accelerate the application of human-centric image generation in reality.

- Secondly, in terms of concrete implementation, inspired by the low-rank refinement phenomenon, our method proposes a new strategy to leverage the idea of MoE . **The novelty of MoLE is that we leverage a soft assignment mechanism in MoE to adaptively activate customized hand and face experts with suitable weights to refine the corresponding part.**  It is significantly different from previous generation methods leveraging MoE, e.g., ERNIE-ViLG and eDiff-i, which primarily employ different experts to denoise the whole image in respective time stages.


### For dataset

Our human-centric dataset has two aspects of advantages:

- **Firstly, our human-centric dataset is of high quality.** As we mentioned in **A1 For Reviewer B5SW**, our human-centric dataset consists of images of high resolution (basically over 1024 × 2048) and large file size, e.g., 1M, 3M, 5M, 10M, etc collected from websites (e.g., [https://www.pexels.com/search/people/](https://www.pexels.com/search/people/), and [https://unsplash.com/s/photos/people](https://unsplash.com/s/photos/people)). The reviewers can simply click these links to have a see. In contrast, the average height and width of the human-centric images from LAION2b-en are approximately 455 and 415 respectively. Most of them are between 320 and 357. Compared to our human-centric dataset, they fail to provide sufficient and high-quality details.
- **Secondly, our human-centric dataset is comprehensive.** As the face and hand are relatively hard to generate naturally compared to other parts as discussed in the AIGC community, besides the collected human-in-the-scene subset, we also provide two high-quality close-up of face and hand subsets, especially the close-up of hand images. **This is absent in human-centric images from LAION2b-en as mentioned in A1**. Additionally, through meticulous manual cropping, we now have curated 20k high-quality hand close-ups (https://sites.google.com/view/response-close-up-of-hand/homepage). **To the best of our knowledge, this quantity of close-up hand images is absent in prior related studies.** It will significantly help to address challenges associated with generating natural-looking hands and propel the advancement of human-centric image generation for subsequent researchers.

These two advantages, we believe, have significant benefits to help generate high-quality human-centric images and accelerate its application in reality.

[1] Faces in Places: Compound Query Retrieval. BMVC 2016

[2] A Style-Based Generator Architecture for Generative Adversarial Networks. CVPR 2019

---

### Meta-Review · Area_Chair_QDyj · 2023-12-08

**Metareview:**

The submission addresses the challenge of improving human-centric text-to-image generation, specifically focusing on faces and hands. The authors curate a dataset of one million high-quality human images, including close-up facial and hand images, to address the limitations of existing models in generating realistic human features.

They propose a Multi-stage Mixture of Low-rank Experts (MoLE) training framework, leveraging separate training on facial and hand images to enhance image quality. Reviewers acknowledge that the proposed method improves the generation of facial and hand details and reduces artifacts like ghost fingers. Numerical metrics suggest competitive performance compared to other text-to-image models on human-centric generation.


The reviewers acknowledged the improvement the submission achieved but also left several important concerns, including the contribution and novelty of the proposed method, limited performance, and lack of ethical consideration.

The authors put vast efforts into the rebuttal. AC also reads the reviews and responses. Some parts of the reviewers' concerns were well addressed, but unfortunately, the authors failed to convince the reviewers of some key concerns (in view of this AC).

The reviewers left private comments for further discussion among the reviewers and ACs.
The main concerning points include: 1) the detailed analysis and information about the dataset despite the contribution claim, 2) the proposed method is incremental compared to the widespread practices, and 3) limited performance inherited from the base model (even the base model is outdated).


PS: Dear authors, do not use EXTERNAL URLs to present the authors' results that might expose the identity of the authors during the rebuttal response, which may lead to a policy violation.

**Justification For Why Not Higher Score:**

The AC concurs with the reviewers' comments. Although the widespread practices would be considered unpublished work, if those are so common, the authors should have made strong analyses to exhibit clear benefits and understanding (motivation) of the proposed method more systematically, so that the proposed method can be significantly contrasted with those existing practices. Also, if the dataset is a key contribution, the authors should have resolved all the potential ethical concerns and have presented thorough analyses of the dataset like other dataset or benchmark papers.

As admitted to the authors, this work can be further improved. This AC thinks that this work is not ready to be published in the current version, and recommends the authors to improve their work further and submit other conferences (e.g., Dataset track).

**Justification For Why Not Lower Score:**

.

---

### Decision · Program_Chairs · 2024-01-16

Reject